# Metrics for evaluating the "quality" in linear atmospheric inverse problems: a case study of a trace gas inversion

Vineet Yadav[1], Subhomoy Ghosh[2,3], and Charles E. Miller[1]

[1]Jet Propulsion Laboratory, California Institute of Technology, 4800 Oak Grove Drive, Pasadena, CA, USA
[2]University of Notre Dame, Notre Dame, IN, USA
[3]National Institute of Standards and Technology, Gaithersburg, MD, USA

**Correspondence:** Subhomoy Ghosh (sghosh4@nd.edu)

**Abstract.** Several metrics have been proposed and utilized to diagnose the performance of linear Bayesian and geostatistical atmospheric inverse problems. These metrics primarily assess reductions in prior uncertainties, compare modeled observations to true observations, and check distributional assumptions. Although important, these metrics should be augmented with sensitivity analysis to obtain a comprehensive understanding of atmospheric inversion performance and improve the quality and confidence in the inverse estimates. In this study, we derive closed-form expressions of local sensitivities for various inputs, including measurements, covariance parameters, covariates, and a forward operator. To further enhance our understanding, we complement local sensitivity analysis with a framework for global sensitivity analysis that can apportion the uncertainty in inputs to the uncertainty associated with inverse estimates. Additionally, we propose a mathematical framework to construct nonstationary correlation matrices from a pre-computed forward operator, which is closely tied to the overall quality of inverse estimates. We demonstrate the application of our methodology in the context of an atmospheric inverse problem for estimating methane fluxes in Los Angeles, California.

## 1 Introduction

In atmospheric applications, inverse models are frequently used to estimate global to regional scale fluxes of trace gases from atmospheric measurements (Enting, 2002). At a global scale, data assimilation remains the primary inverse modeling framework, which assimilates observations sequentially and updates the prior estimates of fluxes by utilizing an atmospheric model coupled with chemistry (for further details on data assimilation, see Wikle and Berliner, 2007). At a regional scale, inversions that assimilate all observations simultaneously by utilizing a pre-computed forward operator (Lin et al., 2003) that describes the relationship between observations and fluxes are commonly used (for details, see Enting, 2002). This work focuses on the use of pre-computed forward operators for atmospheric inverse modeling and addresses sensitivity analysis and correlation in the forward operator in the context of Bayesian (e.g., Lauvaux et al., 2016) and geostatistical inverse methods (e.g., Kitanidis, 1996).

The sensitivity analysis in this work is covered under local and global themes. Primarily, we focus on local sensitivity analysis (LSA), which measures the effect of a given input on a given output and is obtained by computing partial derivatives of an

output quantity of interest for an input factor (see See Rabitz, 1989, and Turányi, 1990). Within the global theme (designated as Global Sensitivity Analysis), we focus on how uncertainty in the model output can be apportioned to different model inputs (Saltelli et al., 2008).

Overall, in atmospheric trace gas inversions, mostly LSA is performed. Within this context, LSA assesses how sensitive the posterior estimates of fluxes are regarding the underlying choices or assumptions, like (1) observations included, (2) model-data error covariance, (3) the input prior information and its error, and (4) the forward operator (for discussion, see Michalak et al., 2017). This task is sometimes performed to arrive at a robust estimate of fluxes and their uncertainties, by running an inverse model multiple times while varying the inputs and assessing their impact on the estimated fluxes and uncertainties. Another complementary way to do LSA is by computing local partial derivatives of inputs that go into an inversion.

LSA can be grouped with standard information content approaches such as an averaging kernel and degrees of freedom for signal (DOFS; for details, see Sec. 2.2.1 of this manuscript, Rodgers, 2000, and Brasseur and Jacob, 2017). However, LSA is more informative than these approaches alone, as it examines individual components (see Sec. 2.2) that determine DOFS and quantifies the impact and relative importance of various components of an inversion.

In this study, we focus on the quality of the inverse estimates of the fluxes, which means providing diagnostic metrics to improve our understanding of the impact of input choices on the inverse estimates of fluxes and thus improve the quality of the inverse model. Specifically, in this technical note, we provide (1) closed-form expressions to conduct LSA by computing partial derivatives, (2) a scientifically interpretable framework for ranking thousands of spatiotemporally correlated input parameters with the same or different units of measurement, (3) a mathematical schema for conducting global sensitivity analysis (GSA), and (4) a technique to assess the spatiotemporal correlation between forward operators of two or multiple observations, which is tied to the overall diagnostics of the estimated fluxes and can lead to improved representation of errors in the forward operator.

## 2   Methods and derivation

In a generic form, a linear inverse problem can be written as:

$$\mathbf{z} = \mathbf{Hs} + \boldsymbol{\epsilon}, \tag{1}$$

where $\mathbf{H}$ is a forward operator that maps model parameters (fluxes in the context of this work) to measurements $\mathbf{z}$ and encapsulates our understanding of the physics of the measurements. The error $\boldsymbol{\epsilon}$ in Eq. (1) describes the mismatch between measurements and the modeled measurements (see Sec. 3).

In a typical linear atmospheric inverse problem (see Fig. 1), the estimates of fluxes (box 8 of Fig. 1) are obtained in a
classical one-stage batch Bayesian setup (for details, see Enting, 2002; Tarantola, 2005). In this setup, the a priori term (box 3
in Fig. 1) is based on a fixed flux pattern, and errors (box 6 in Fig. 1) are either assumed to be independent or are governed by
a pre-defined covariance structure (for details, see Gurney et al., 2003; Rödenbeck et al., 2003, 2006).

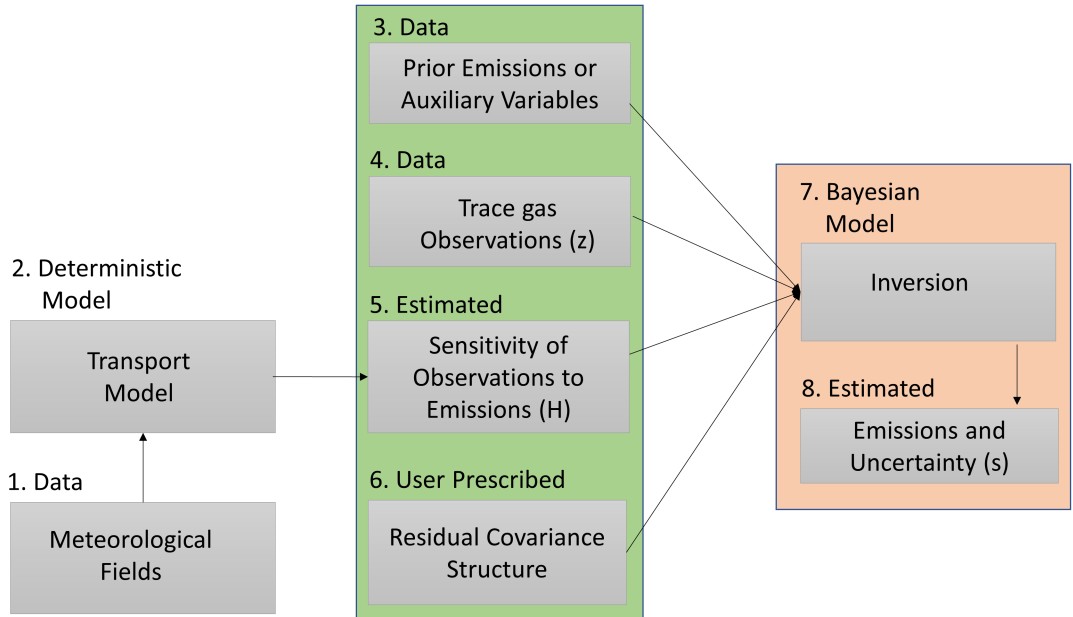

**Figure 1.** The schema for performing a linear atmospheric inversion to obtain estimates of the fluxes of greenhouse gases. The middle column (the green background box) lists all the inputs that are required for performing an inversion whereas the right column (the orange background box) lists the modeling process (box 7) and the output obtained after performing an inversion (box 8). Note this work focuses on understanding and ranking the impact of the inputs (box 3, 4, and 6 in the middle column) on the estimates of fluxes (box 8) and developing correlation structures from the forward operator (box 5).

Within the previously mentioned setup, the choice of the input parameters, including the forms of error structures, pro-
foundly impacts the quality of the inverse estimates of fluxes. Understanding the impact of these inputs is critical for evaluating
the quality of the estimated fluxes. Thus, first (Sec. 2.1), we utilize the understanding of the physics of the measurements,
encapsulated in $\mathbf{H}$, to generate scientifically interpretable correlation matrices (box 6 in Fig. 1). Second, we assess and rank
the importance of the inputs (Sec. 2.2) shown in the middle column (the green background box) of Fig. 1 (box 8 of Fig. 1),
which is finally followed, by methane ($CH_4$) case study that demonstrates the applicability of our methods (see Sec. 2).

## 2.1 Analysis of the forward operator

In inversions that assimilate all observations simultaneously, a forward operator for each observation included in an inversion is obtained from a transport model. These observations can be obtained from multiple platforms, including an in-situ network of fixed locations on the surface, intermittent aircraft flights, and satellites. In most situations, the spatiotemporal coverage of these forward operators is visually assessed by plotting an aggregated sum or mean of their values over a spatial domain. However, standard quantitative metrics to evaluate their coverage and intensity in space and time remain absent. In this study, we present two metrics for this assessment, which are defined below. These metrics conform to triangular inequality and are distances in their respective metric spaces.

Note that sometimes in the published literature on trace gas inversions, the forward operator obtained from a transport model is referred to as a sensitivity matrix, Jacobian, or footprint. Henceforth, we always refer to the Jacobian/sensitivity matrix or footprint as a forward operator to avoid misinterpretation. We show our application through forward operators constructed by running a Lagrangian transport model. However, the proposed methods can also be applied in the Eulerian framework (see Brasseur and Jacob, 2017 for details).

### 2.1.1 Integrated area overlap measurement index (IAOMI)

The Integrated Area Overlap Measurement Index (IAOMI) summarizes the shared information content between two forward operators and hence indirectly between two observations. It is, therefore, a measure of the uniqueness of the flux signal associated with an observation compared to other observations.

Intuitively, IAOMI can be better understood spatially. For a given time point, consider two forward operators $\mathbf{F}$ and $\mathbf{G}$ as two vector-valued functions over an area. Index IAOMI is the proportion of the common contribution of the two forward operators from the intersected area with respect to the overall contribution of the two forward operators. This is demonstrated through a Venn diagram in Fig. 2. Thus, IAOMI can be defined as:

$$\nu_{\mathbf{F},\mathbf{G}} = \frac{\sum_{A_{\mathbf{F}} \cap A_{\mathbf{G}}} \mathbf{f}_1(\mathbf{F},\mathbf{G})}{\sum_{A_{\mathbf{F}} \cup A_{\mathbf{G}}} \mathbf{f}_2(\mathbf{F},\mathbf{G})}, \tag{2}$$

where for any forward operator $\mathbf{S}$, the corresponding set $A_{\mathbf{S}}$ on which forward operator is always positive, is defined as $A_{\mathbf{S}} = \{\boldsymbol{x} : \mathbf{S}(\boldsymbol{x}) > 0\}$ and the two vector-valued functionals $\mathbf{f}_1$ and $\mathbf{f}_2$ can be given as:

$$\mathbf{f}_1(\mathbf{F},\mathbf{G}) = \begin{cases} min(\mathbf{F},\mathbf{G}) & \text{on } A_{\mathbf{F}} \cap A_{\mathbf{G}} \\ 0 & \text{otherwise} \end{cases} \quad \text{and} \quad \mathbf{f}_2(\mathbf{F},\mathbf{G}) = \begin{cases} max(\mathbf{F},\mathbf{G}) & \text{on } A_{\mathbf{F}} \cap A_{\mathbf{G}} \\ \mathbf{F} & \text{on } A_{\mathbf{F}} \cap A_{\mathbf{G}}^c \\ \mathbf{G} & \text{on } A_{\mathbf{F}}^c \cap A_{\mathbf{G}} \end{cases} \tag{3}$$

Note that the IAOMI defined in Eq. (2) can also be written as a ratio of the sum of minimums over sum of the maximums as:

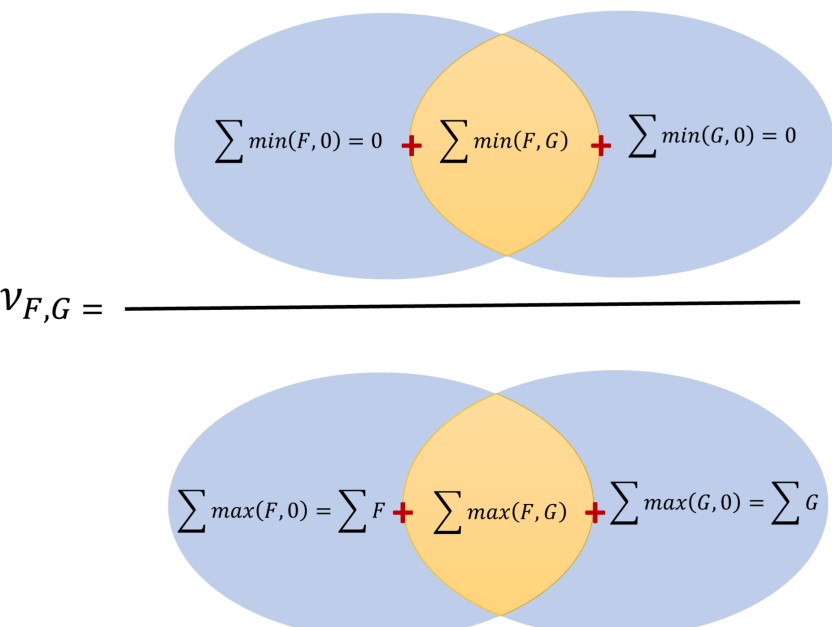

**Figure 2.** Venn diagram that defines IAOMI in terms of two hypothetical forward operators $\mathbf{F}$ and $\mathbf{G}$

$$\nu_{\mathbf{F},\mathbf{G}} = \frac{\Sigma_{A_{\mathbf{F}} \cup A_{\mathbf{G}}} min(\mathbf{F},\mathbf{G})}{\Sigma_{A_{\mathbf{F}} \cup A_{\mathbf{G}}} max(\mathbf{F},\mathbf{G})} \qquad (4)$$
IAOMI $\nu$ can also be thought as a measure of similarity between two forward operators. It is evident from Eq. (4) that this is
a weighted Jaccard similarity index or Ruzicka index (Cha, 2007) which describes similarity between two forward operators
$\mathbf{F}$ and $\mathbf{G}$. It follows that $\nu$ is closed and bounded in $[0,1]$ and accounts for both the spatiotemporal spread and the intensity of
the forward operator. A stronger $\nu$ implies larger overlap of intensity in space and time, is analogous to finding the common
area within two curves, and is indicative of the magnitude of overlapping information, a knowledge beneficial in the context of
satellite observations with a higher potential for sharing information content.

A measure of dissimilarity can be obtained from $\nu$ and can be defined by $1-\nu$. The smaller the overlap or the larger the value
of $1-\nu$, the more significant the disparity. Note the $\nu$ metric is only indicative of the overlap in the spatiotemporal intensity
between two forward operators. To measure how much of the shared intensity has come from either forward operator, we use
a metric $\upsilon_{\mathbf{F}|(\mathbf{F},\mathbf{G})}$ defined as:
$$\upsilon_{\mathbf{F}|(\mathbf{F},\mathbf{G})} = \frac{\Sigma_{A_{\mathbf{F}} \cap A_{\mathbf{G}}} \mathbf{f}_1(\mathbf{F},\mathbf{G})}{\Sigma_{A_{\mathbf{F}}} \mathbf{f}_3(\mathbf{F})}, \qquad (5)$$
where $\mathbf{f}_3(\mathbf{F}) = F$ on $A_\mathbf{F}$ and 0 everywhere else. Likewise, we can define $\upsilon_{\mathbf{G}|(\mathbf{F},\mathbf{G})}$ which shows proportional contribution of
the forward operator G on the shared intensity. Both $\nu$ and $\upsilon$ can be computed from observations taken from same or different
platforms, at same or different time or for two different in-situ measurement sites over a specified time-interval.

### 2.1.2 Spatio-temporal Area of Dominance (STAD)

The spatiotemporal area of dominance (STAD) stems naturally from IAOMI. For any two forward operators $\mathbf{F}$, and $\mathbf{G}$, we can
find out the left-over dominant contribution of $\mathbf{F}$ and $\mathbf{G}$ by computing quantities $\mathbf{F} - \mathbf{G}$ and $\mathbf{G} - \mathbf{F}$ that lead to the determina-
tion of the areas where $\mathbf{F}$ or $\mathbf{G}$ is dominant.

For two forward operators $\mathbf{F}$ and $\mathbf{G}$, STAD of $\mathbf{F}$ with respect to $\mathbf{G}$ is defined as:
$$
\text{STAD}_\mathbf{F}(\mathbf{F},\mathbf{G}) = \begin{cases} \mathbf{F} - min(\mathbf{F},\mathbf{G}) & \text{on } A_\mathbf{F} \cap A_\mathbf{G} \\ \mathbf{F} & \text{otherwise} \end{cases}
$$

IAOMI and STAD of any forward operator $\mathbf{F}$ with respect to the forward operators $\mathbf{F}$ and $\mathbf{G}$ are linked by the following
equation:
$$
\nu_{\mathbf{F},\mathbf{G}} \Sigma_{A_\mathbf{F} \cup A_\mathbf{G}} H_2(\mathbf{F},\mathbf{G}) + \Sigma_{A_\mathbf{F} \cup A_\mathbf{G}} STAD_\mathbf{F}(\mathbf{F},\mathbf{G}) = \Sigma_{A_\mathbf{F}} \mathbf{F} \tag{6}
$$

Given a number of forward operators $\{\mathbf{F}, \mathbf{G}_1, \mathbf{G}_2, \cdots\}$, STAD for any particular forward operator $\mathbf{F}$ with respect to all other
forward operators can be generalized from Eq. (6) as $\mathbf{F}_{\text{STAD}}(\mathbf{F}, \mathbf{G}_{\text{max}})$ where $\mathbf{G}_{\text{max}} = \max_i \mathbf{G}_i$ on $A_\mathbf{G}$; $A_\mathbf{G} = \cup_k A_{\mathbf{G}_k}$ and
$A_{\mathbf{G}_k}$ is the set on which forward operator $\mathbf{G}_k$ is always positive (see Sec. 2.1.1 for its definition). STAD can be aggregated
over any time-periods. Intuitively, STAD determines areas in space-time where one forward operator dominates over other
forward operators, which is especially useful in locating the primary flux sources that influence an observation.

One can use 1-IAOMI or distance metric like Jensen-Shannon distance (JSD; see Appendix B) matrix of all pairwise forward
operators as a representative distance matrix for describing correlations in model-data errors (i.e., $\mathbf{R}$ in Eq. (7)). As JSD or
1-IAOMI matrices are real, symmetric, and admit orthogonal decomposition, the entry-wise exponential of such symmetric
diagonalizable matrices is positive-semidefinite and can be incorporated in model data mismatch matrix $\mathbf{R}$ (see Ghosh et al.,
2021). Furthermore, the IAOMI matrix itself is a positive semidefinite (Bouchard et al., 2013) matrix and can also be directly
incorporated in $\mathbf{R}$ as a measure of correlation. This is an example of how IAOMI or 1- IAOMI could be particularly useful for
satellite data based inversions with higher degree of spatial overlap of the forward operators. However, we do not explore this
area of research in this manuscript.

## 2.2 Local sensitivity analysis (LSA) in inversions

For linear Bayesian and geostatistical inverse problem, the solutions (see, Tarantola, 2005 for the batch Bayesian and Kitanidis, 1996 for the geostatistical case) can be obtained by minimizing their respective objective functions. These objective functions can be given as:

$$L(\mathbf{s}|\mathbf{y}, \mathbf{s}_{\text{prior}}, \mathbf{H}, \mathbf{Q}, \mathbf{R}) = \frac{1}{2}(\mathbf{z} - \mathbf{Hs})^t \mathbf{R}^{-1}(\mathbf{z} - \mathbf{Hs}) + \frac{1}{2}(\mathbf{s} - \mathbf{s}_{\text{prior}})^t \mathbf{Q}^{-1}(\mathbf{s} - \mathbf{s}_{\text{prior}}) \tag{7}$$

$$L(\mathbf{s}|\mathbf{y}, \mathbf{H}, \mathbf{Q}, \mathbf{R}, \boldsymbol{\beta}) = \frac{1}{2}(\mathbf{z} - \mathbf{Hs})^t \mathbf{R}^{-1}(\mathbf{z} - \mathbf{Hs}) + \frac{1}{2}(\mathbf{s} - \mathbf{X}\boldsymbol{\beta})^t \mathbf{Q}^{-1}(\mathbf{s} - \mathbf{X}\boldsymbol{\beta}), \tag{8}$$

where lowercase symbols represent vectors and the uppercase symbols represent matrices, and this exact representation is adopted throughout the manuscript. In Eq. (7) and (8), $\mathbf{z}$ is an $(n \times 1)$ vector of available measurements with unit of each entry being ppm. The forward operator $\mathbf{H}$ is an $(n \times m)$ matrix with unit of each entry being ppm $\mu$moles$^{-1}$m$^2$sec. The matrix $\mathbf{H}$ is obtained from a transport model that describes the relationship between measurements and unknown fluxes. Unknown flux $\mathbf{s}$ is an $(m \times 1)$ vector with unit of entries being $\mu$moles m$^{-2}$sec$^{-1}$. The covariance matrix $\mathbf{R}$ of the model-data errors is an $(n \times n)$ matrix with unit of the entries being ppm$^2$. The covariate matrix $\mathbf{X}$ is an $(m \times p)$ matrix of known covariates related to $\mathbf{s}$. The unit of each of the entries in every column of the covariate matrix $\mathbf{X}$ is the unit of its measurement or if it is standardized (e.g. subtract the mean from the covariate and divide by its standard deviation) then it is unitless. For further discussion on standardization and normalization see Gelman and Hill, 2006. The units of $(p \times 1)$ vector $\boldsymbol{\beta}$ are such that $\mathbf{X}\boldsymbol{\beta}$ and $\mathbf{s}$ have the same units. The prior error covariance matrix $\mathbf{Q}$ is an $(m \times m)$ matrix that represents the errors between $\mathbf{s}$ and $\mathbf{X}\boldsymbol{\beta}$ with unit of the entries being $\left(\mu\text{moles m}^{-2}\text{sec}^{-1}\right)^2$.

The analytical solutions for the unknown fluxes $\mathbf{s}$ in the Bayesian case (denoted by the subscript B) and the geostatistical case (denoted by the subscript G) can be obtained from Eq. (9) and (10) as given below.

$$\hat{\mathbf{s}}_B = \mathbf{s}_{\text{prior}} + \mathbf{QH}^t \left(\mathbf{HQH}^t + \mathbf{R}\right)^{-1} (\mathbf{z} - \mathbf{Hs}_{\text{prior}}) \tag{9}$$

$$\hat{\mathbf{s}}_G = \mathbf{X}\boldsymbol{\beta} + \mathbf{QH}^t \left(\mathbf{HQH}^t + \mathbf{R}\right)^{-1} (\mathbf{z} - \mathbf{HX}\boldsymbol{\beta}) \tag{10}$$

In linear Bayesian and geostatistical inverse problems described by equations 7 and 8, the estimated fluxes can be expressed as the sum of the prior information and the update obtained from the observations. In equations 9 and 10, the second term represents the observational constraint, while the first term describes the prior information (in Eq. 9) and the information about fluxes (through $\mathbf{X}$ in Eq. 10). When there is no additional information, the solution corresponds to the prior knowledge. Since the estimate of $\mathbf{s}_G$ in Eq. (10) depends on the unknown $\boldsymbol{\beta}$, it requires prior estimation of $\boldsymbol{\beta}$ before obtaining $\hat{\mathbf{s}}_G$. The solution for the $\hat{\boldsymbol{\beta}}$ can be obtained from pre-determined quantities as described earlier in the context of Eq. (8) and can be given as:

$$\hat{\boldsymbol{\beta}} = \boldsymbol{\Omega}^{-1} \mathbf{A}^t \boldsymbol{\Psi}^{-1} \mathbf{z}, \tag{11}$$

plugging in $\hat{\boldsymbol{\beta}}$ in Eq. (10) leads to Eq. (12) where all symbols are defined previously or in Eq. (13).
$$\hat{\mathbf{s}}_G = \mathbf{X}\boldsymbol{\Omega}^{-1}\mathbf{A}^t\boldsymbol{\Psi}^{-1}\mathbf{z} + \mathbf{Q}\mathbf{H}^t\boldsymbol{\Psi}^{-1}\left(\mathbf{z} - \mathbf{A}\boldsymbol{\Omega}^{-1}\mathbf{A}^t\boldsymbol{\Psi}^{-1}\mathbf{z}\right), \quad \text{where} \tag{12}$$
$$\mathbf{A} = \mathbf{H}\mathbf{X}, \boldsymbol{\Psi} = \left(\mathbf{H}\mathbf{Q}\mathbf{H}^t + \mathbf{R}\right), \boldsymbol{\Omega} = (\mathbf{H}\mathbf{X})^t \left(\mathbf{H}\mathbf{Q}\mathbf{H}^t + \mathbf{R}\right)^{-1}\mathbf{H}\mathbf{X} \tag{13}$$
Note that, $\hat{\mathbf{s}}_B$ and $\hat{\mathbf{s}}_G$ in Eq. (9) and (10) are essentially functions that are represented by equations. It is a commonly adopted
nomenclature that is used by researchers working in the field of atmospheric inversions. We differentiate Eq. (9) with respect
to $\mathbf{s}_{\text{prior}}$, $\mathbf{R}$, $\mathbf{Q}$, $\mathbf{z}$ and Eq. (12) with respect to $\mathbf{X}$, $\mathbf{R}$, $\mathbf{Q}$, $\mathbf{z}$ to obtain the local sensitivities. There are two ways to differentiate $\hat{\mathbf{s}}$
with respect to $\mathbf{z}$, $\mathbf{X}$, $\mathbf{H}$, $\mathbf{Q}$, and $\mathbf{R}$. In the first case, every entry in $\mathbf{z}$, $\mathbf{X}$, $\mathbf{H}$, $\mathbf{Q}$, and $\mathbf{R}$ can be considered as a parameter that
results in differentiation of $\hat{\mathbf{s}}$ with respect to these quantities. An "entry" refers to each element of the matrix denoted by $ij$,
where $i$ represents the row number and $j$ represents the column number. On the other hand, if the structures of the covariance
matrices $\mathbf{Q}$ and $\mathbf{R}$ are determined by parameters then $\hat{\mathbf{s}}$ can be differentiated just with respect to these parameters. In the former
case, Eq. (9) and (12) are used to differentiate $\hat{\mathbf{s}}$ with respect to an entry at a time in $\mathbf{z}$, $\mathbf{X}$, $\mathbf{H}$, $\mathbf{Q}$, and $\mathbf{R}$. Such an approach of
entry-by-entry differentiation is useful if the computational cost in terms of memory constraint is important or if we would like
to know the influence of a single entry on $\hat{\mathbf{s}}$. We provide both sets of equations in this manuscript.
**2.2.1   LSA with respect to observations, priors, scaling factors, and forward operators**
Local sensitivity of $\hat{\mathbf{s}}$ with respect to observations ($\mathbf{z}$) can be given as:
$$\frac{\partial \hat{\mathbf{s}}_B}{\partial \mathbf{z}} = \mathbf{Q}\mathbf{H}^t\boldsymbol{\Psi}^{-1} \tag{14}$$
$$\frac{\partial \hat{\mathbf{s}}_G}{\partial \mathbf{z}} = \mathbf{X}\boldsymbol{\Omega}^{-1}\mathbf{A}^t\boldsymbol{\Psi}^{-1} + \mathbf{Q}\mathbf{H}^t\boldsymbol{\Psi}^{-1} - \mathbf{Q}\mathbf{H}^t\boldsymbol{\Psi}^{-1}\mathbf{A}\boldsymbol{\Omega}^{-1}\mathbf{A}^t\boldsymbol{\Psi}^{-1}, \tag{15}$$
where all quantities are as defined earlier. The units of the entries in $\frac{\partial \hat{\mathbf{s}}}{\partial \mathbf{z}}$ are $\mu\text{moles}^{-1}\text{m}^2\text{sec}^{-1}\text{ppm}^{-1}$ and the matrices are of
dimension $(m \times n)$. These units are inverse of the units of $\mathbf{H}$. Local sensitivities with respect to an observation $z_i$ for both the
Bayesian and the geostatistical case can be written as a vector of sensitivities times an indicator for the $i^{\text{th}}$ entry i.e. $\frac{\partial \hat{\mathbf{s}}}{\partial \mathbf{z}}\mathbf{e}_i$ where
$\mathbf{e}_i = \frac{\partial \mathbf{z}}{\partial z_i}$ is a vector of zeros with the $i^{\text{th}}$ entry equal to 1.

Note by utilizing $\frac{\partial \hat{\mathbf{s}}}{\partial \mathbf{z}}$, we can also obtain an averaging kernel (or model resolution matrix) and DOFS (see Rodgers, 2000).
The averaging kernel matrix for any linear inverse model can be written as:
$$\mathbf{V} = \frac{\partial \hat{\mathbf{s}}}{\partial \mathbf{z}} \times \mathbf{H}, \tag{16}$$
where $\mathbf{V}$ of dimension $(m \times m)$ is the local sensitivity of $\hat{\mathbf{s}}$ with respect to the true unknown fluxes. Then the DOFS can be
computed by taking the trace of the averaging kernel matrix $\mathbf{V}$. DOFS represents the amount of information resolved by an

inverse model when a set of observations have been assimilated (for a detailed discussion, see Rodgers, 2000 and Brasseur and Jacob, 2017). Theoretically, the value of DOFS cannot exceed the number of observations ($n$) in an underdetermined system and the number of fluxes ($m$) in an overdetermined system.

We can directly compute local sensitivity of $\hat{s}$ with respect to the prior mean flux $\mathbf{s}_{\text{prior}}$ in the Bayesian case. In the geostatistical case, the prior mean is modeled by two quantities $\mathbf{X}$ and $\boldsymbol{\beta}$. In this scenario, we need to find sensitivities with respect to $\mathbf{X}$ as well as $\boldsymbol{\beta}$. These local sensitivities can be given as:

$$\frac{\partial \hat{\mathbf{s}}_B}{\partial \mathbf{s}_{\text{prior}}} = \mathbf{I} - \mathbf{CH} \tag{17}$$

$$\frac{\partial \hat{\mathbf{s}}_G}{\partial \mathbf{X}} = \mathbf{K}_z \otimes \left(\mathbf{I} + \left(\mathbf{MA}^t - \mathbf{X}\boldsymbol{\Omega}^{-1}\mathbf{A}^t - \mathbf{QH}^t\right)\boldsymbol{\Psi}^{-1}\mathbf{H}\right) + \left(\mathbf{X}\boldsymbol{\Omega}^{-1} - \mathbf{M}\right) \otimes \left(\mathbf{F}_z - \mathbf{K}_z\mathbf{A}^t\boldsymbol{\Psi}^{-1}\mathbf{H}\right) \tag{18}$$

$$\frac{\partial \hat{\mathbf{s}}_G}{\partial \hat{\boldsymbol{\beta}}} = \mathbf{X} - \mathbf{CA}, \tag{19}$$

where $\mathbf{A} = \mathbf{HX}, \mathbf{B} = \mathbf{QH}^t, \mathbf{C} = \mathbf{B}\boldsymbol{\Psi}^{-1}, \boldsymbol{\Omega} = \mathbf{A}^t\boldsymbol{\Psi}^{-1}\mathbf{A}, \mathbf{K}_z = \mathbf{z}^t\boldsymbol{\Psi}^{-1}\mathbf{A}\boldsymbol{\Omega}^{-1}, \mathbf{M} = \mathbf{CA}\boldsymbol{\Omega}^{-1}$, and $\mathbf{F}_z = \mathbf{z}^t\boldsymbol{\Psi}^{-1}\mathbf{H}$. The symbol $\otimes$ represents the Kronecker product. The quantity $\frac{\partial \hat{\mathbf{s}}_B}{\partial \mathbf{s}_{\text{prior}}}$ is of dimension ($m \times m$) and its entries are unitless. The quantity $\frac{\partial \hat{\mathbf{s}}_G}{\partial \hat{\boldsymbol{\beta}}}$ is of dimension ($m \times p$) and units of the entries in each column of $\frac{\partial \hat{\mathbf{s}}_G}{\partial \hat{\boldsymbol{\beta}}}$ are of the form ($\mu$moles$^{-1}$m$^2$sec$^{-1}$)(unit of $\beta_i$)$^{-1}$. The sensitivity matrix $\frac{\partial \hat{\mathbf{s}}_G}{\partial \mathbf{X}}$ is of dimension ($m \times mp$) where every $i^{\text{th}}$ block of $m$ columns ($(i-1)m + A : im$) of $\frac{\partial \hat{\mathbf{s}}_G}{\partial \mathbf{X}}$ has units of the form ($\mu$moles$^{-1}$m$^2$sec$^{-1}$)(unit of $\mathbf{X}_i$)$^{-1}$ where $\mathbf{X}_i$ is the $i^{\text{th}}$ column of $\mathbf{X}$. Note that the sensitivity matrix $\frac{\partial \hat{\mathbf{s}}_B}{\partial \mathbf{s}_{\text{prior}}}$ in Eq. (17) can also be considered as a proportion of posterior uncertainty to that of the prior uncertainty. In context of the Bayesian case, proportional uncertainty reduction becomes averaging kernel.

Sometimes, it is essential to know the influence of the prior of any particular grid point or an area consisting of few grid-cells within $\hat{s}$. Local sensitivity of $\hat{s}$ with respect to the $i^{\text{th}}$ entry in $\mathbf{s}_{\text{prior}}$ and $\hat{\beta}_i$ is a matrix of dimension ($m \times 1$) and can be written as $\frac{\partial \hat{\mathbf{s}}_B}{\partial \mathbf{s}_{\text{prior}}}\mathbf{e}_i$ and $\frac{\partial \hat{\mathbf{s}}_G}{\partial \hat{\boldsymbol{\beta}}}\mathbf{e}_i$ respectively. However, the entry-wise $\frac{\partial \hat{\mathbf{s}}_G}{\partial \mathbf{X}_{ij}}$ is more complex and can be given by:

$$\frac{\partial \hat{\mathbf{s}}_G}{\partial X_{ij}} = (\mathbf{I} - \mathbf{CH}) \left( \left(\mathbf{I} - \mathbf{X}\boldsymbol{\Omega}^{-1}\mathbf{X}^t\mathbf{H}^t\boldsymbol{\Psi}^{-1}\mathbf{H}\right) \frac{\partial \mathbf{X}}{\partial X_{ij}}\boldsymbol{\Omega}^{-1}\mathbf{X}^t + \mathbf{X}\boldsymbol{\Omega}^{-1}\frac{\partial \mathbf{X}^t}{\partial X_{ij}}\left(\mathbf{I} - \mathbf{H}^t\boldsymbol{\Psi}^{-1}\mathbf{HX}\boldsymbol{\Omega}^{-1}\mathbf{X}^t\right) \right) \mathbf{F}_z^t, \tag{20}$$

where $\frac{\partial \mathbf{X}^t}{\partial X_{ij}} = \mathbf{E}_{ij}$ is a single-entry matrix with a one for a $X_{ij}$ for which differentiation is being performed and zero everywhere else. For $\mathbf{z}$, entry-by-entry differentiation can be easily performed since both Eq. (9) and (12) result from linear models and are functions of the form $\boldsymbol{\Phi}\mathbf{z} + \mathbf{n}$ where $\boldsymbol{\Phi}$ and $\mathbf{n}$ are independent of $\mathbf{z}$. For example, $\boldsymbol{\Phi}$ and $\mathbf{n}$ for Eq. (9) are $\mathbf{QH}^t\left(\mathbf{HQH}^t + \mathbf{R}\right)^{-1}$ and $\mathbf{s}_{\text{prior}} - \mathbf{QH}^t\left(\mathbf{HQH}^t + \mathbf{R}\right)^{-1}\mathbf{Hs}_{\text{prior}}$ respectively and are independent of $\mathbf{z}$. In this case, $\frac{\partial \hat{\mathbf{s}}_B}{\partial z_i}$ can be written as $\boldsymbol{\Phi}\mathbf{e_i}$ where $\mathbf{e_i}$ is a single-entry vector with a one for a $z_i$ for which differentiation is being performed and zero everywhere else. Local sensitivity $\frac{\partial \hat{\mathbf{s}}_G}{\partial z_i}$ can similarly be defined for the respective $\boldsymbol{\Phi}$. Here both the quantities $\frac{\partial \hat{\mathbf{s}}_G}{\partial X_{ij}}$ and $\frac{\partial \hat{\mathbf{s}}_B}{\partial z_i}$ are matrices of dimension ($m \times 1$).

Local sensitivity of $\hat{\mathbf{s}}$ with respect to an entry in the forward operator has units of the form $\left(\mu\text{moles}^{-1}\text{m}^2\text{sec}^{-1}\right)^2\text{ppm}^{-1}$. In
the Bayesian case, this sensitivity can be written as:
$$\frac{\partial\hat{\mathbf{s}}_B}{\partial\mathbf{H}} = \mathbf{Q}\otimes\mathbf{P}_z - \mathbf{B}\mathbf{P}_z\otimes\mathbf{C}^t - \mathbf{B}\mathbf{C}^t\otimes\mathbf{P}_z - \mathbf{Q}\otimes\mathbf{D} + \mathbf{B}\mathbf{D}\otimes\mathbf{C}^t + \mathbf{B}\mathbf{C}^t\otimes\mathbf{D} - \mathbf{s}_{\text{prior}}\otimes\mathbf{C}^t, \tag{21}$$
where $\frac{\partial\hat{\mathbf{s}}_B}{\partial\mathbf{H}}$ is a sensitivity matrix of dimension $(m\times mn)$. In the geostatistical case, this sensitivity can be partitioned into two
components i.e., $\frac{\partial\hat{\boldsymbol{\beta}}}{\partial\mathbf{H}}$ and $\frac{\partial\hat{\boldsymbol{\epsilon}}}{\partial\mathbf{H}}$ as shown in Eq. (22) where $\frac{\partial\hat{\boldsymbol{\beta}}}{\partial\mathbf{H}}$ and $\frac{\partial\hat{\boldsymbol{\epsilon}}}{\partial\mathbf{H}}$ are obtained in an orderly sequence from Eq. (23) and

225    (24).

$$\frac{\partial\hat{\mathbf{s}}_G}{\partial\mathbf{H}} = \mathbf{X}\frac{\partial\hat{\boldsymbol{\beta}}}{\partial\mathbf{H}} + \frac{\partial\hat{\boldsymbol{\epsilon}}}{\partial\mathbf{H}}\quad\text{where} \tag{22}$$
$$\frac{\partial\hat{\boldsymbol{\beta}}}{\partial\mathbf{H}} = -\mathbf{L}\otimes\mathbf{G}_z - \mathbf{P}_z^t\mathbf{A}\boldsymbol{\Omega}^{-1}\mathbf{X}^t\otimes\mathbf{K}^T + \mathbf{G}_z\mathbf{H}\mathbf{Q}\otimes\mathbf{K}^t + \mathbf{N}\otimes\mathbf{G}_z + \mathbf{L}\otimes\mathbf{P}_z^T - \mathbf{P}_z^T\mathbf{H}\mathbf{Q}\otimes\mathbf{K}^t - \mathbf{N}\otimes\mathbf{P}_z^t \tag{23}$$
$$\frac{\partial\hat{\boldsymbol{\epsilon}}}{\partial\mathbf{H}} = \mathbf{Q}\otimes\mathbf{P}_z - \mathbf{C}\mathbf{z}\otimes\mathbf{C}^t - \mathbf{C}\mathbf{H}\mathbf{Q}\otimes\mathbf{P}_z - \mathbf{X}\mathbf{K}^t\mathbf{z}\otimes\mathbf{C}^T - \mathbf{C}\mathbf{A}\frac{\partial\hat{\boldsymbol{\beta}}}{\partial\mathbf{H}} \tag{24}$$
The expanded form of some of the symbols in Eq. (21) through (24), which have not been expanded yet can be written
as $\mathbf{D} = \boldsymbol{\Psi}\mathbf{H}\mathbf{s}_{\text{prior}}$, $\mathbf{G}_z = \mathbf{z}^t\boldsymbol{\Psi}^{-1}\mathbf{A}\boldsymbol{\Omega}^{-1}\mathbf{A}^t\boldsymbol{\Psi}^{-1}$, $\mathbf{L} = \boldsymbol{\Omega}^{-1}\mathbf{X}^t$, $\mathbf{N} = \boldsymbol{\Omega}^{-1}\mathbf{A}^t\boldsymbol{\Psi}^{-1}\mathbf{H}\mathbf{Q}$, $\mathbf{P}_z = \boldsymbol{\Psi}^{-1}\mathbf{z}$, and $\mathbf{K} = \boldsymbol{\Psi}^{-1}\mathbf{A}\boldsymbol{\Omega}^{-1}$. The
quantities $\frac{\partial\hat{\mathbf{s}}_G}{\partial\mathbf{H}}$, $\frac{\partial\hat{\boldsymbol{\beta}}}{\partial\mathbf{H}}$, and $\frac{\partial\hat{\boldsymbol{\epsilon}}}{\partial\mathbf{H}}$ are sensitivity matrices of dimensions $(m\times mn)$, $(p\times mn)$, and $(m\times mn)$ respectively. The units
of the entries of $\frac{\partial\hat{\mathbf{s}}}{\partial\mathbf{H}}$ are of the form $(\mu\text{moles}^{-1}\text{m}^2\text{sec}^{-1})^2\text{ppm}^{-1}$.

There might be times when we would like to know the sensitivity of the transport ($\mathbf{H}$) with respect to certain source locations
only. In this case, we can use $ij$ form of Eq. (21) through (24) to obtain $\frac{\partial\hat{\mathbf{s}}_B}{\partial H_{ij}}$ in parts. In this formulation, $\frac{\partial\hat{\mathbf{s}}_B}{\partial H_{ij}}$ can be given
as:
$$\frac{\partial\hat{\mathbf{s}}_B}{\partial H_{ij}} = \mathbf{C}\frac{\partial\mathbf{H}}{\partial H_{ij}}\left(\mathbf{C}(\mathbf{H}\mathbf{s}_{\text{prior}} - \mathbf{z}) - \mathbf{s}_{\text{prior}}\right) + (\mathbf{Q} - \mathbf{C}\mathbf{H}\mathbf{Q})\left(\frac{\partial\mathbf{H}}{\partial H_{ij}}\right)^t\boldsymbol{\Psi}^{-1}(\mathbf{z} - \mathbf{H}\mathbf{s}_{\text{prior}}) \tag{25}$$
$$\frac{\partial\hat{\mathbf{s}}_G}{\partial H_{ij}} = \mathbf{X}\frac{\partial\hat{\boldsymbol{\beta}}}{\partial H_{ij}} + \frac{\partial\hat{\boldsymbol{\epsilon}}}{\partial H_{ij}},\quad\text{where} \tag{26}$$
$$\frac{\partial\hat{\boldsymbol{\beta}}}{\partial H_{ij}} = \left(-\mathbf{K}^t\frac{\partial\mathbf{H}}{\partial H_{ij}}\left(\mathbf{X}\mathbf{N} - \mathbf{C}\mathbf{A}\mathbf{S} + \mathbf{Q}\mathbf{H}^t\right) + \mathbf{K}^t\mathbf{H}\mathbf{Q}\frac{\partial\mathbf{H}^t}{\partial H_{ij}}\left(\boldsymbol{\Psi}^{-1}\mathbf{A}\mathbf{S}^t - \mathbf{I}\right) + \boldsymbol{\Omega}^{-1}\mathbf{X}^t\frac{\partial\mathbf{H}^t}{\partial H_{ij}}\left(\mathbf{I} - \boldsymbol{\Psi}^{-1}\mathbf{A}\mathbf{S}\right)\right)\boldsymbol{\Psi}^{-1}\mathbf{z} \tag{27}$$
$$\frac{\partial\hat{\boldsymbol{\epsilon}}}{\partial H_{ij}} = \left(\mathbf{Q}\frac{\partial\mathbf{H}^t}{\partial H_{ij}} - \mathbf{C}\frac{\partial\mathbf{H}}{\partial H_{ij}}\mathbf{Q}\mathbf{H}^t - \mathbf{C}\mathbf{H}\mathbf{Q}\frac{\partial\mathbf{H}^t}{\partial H_{ij}}\right)\boldsymbol{\Psi}^{-1}\left(\mathbf{z} - \mathbf{A}\hat{\boldsymbol{\beta}}\right) - \mathbf{C}\left(\frac{\partial\mathbf{H}}{\partial H_{ij}}\mathbf{X}\hat{\boldsymbol{\beta}} + \mathbf{A}\frac{\partial\hat{\boldsymbol{\beta}}}{\partial H_{ij}}\right), \tag{28}$$
where $\mathbf{S} = \mathbf{A}\boldsymbol{\Omega}^{-1}$ and the matrix $\frac{\partial\mathbf{H}}{\partial H_{ij}}$ is a single-entry matrix with a one for a $H_{ij}$ entry for which the differentiation is being
performed and zero everywhere else. The quantities $\frac{\partial\hat{\mathbf{s}}_B}{\partial H_{ij}}$, $\frac{\partial\hat{\mathbf{s}}_G}{\partial H_{ij}}$, $\frac{\partial\hat{\boldsymbol{\beta}}}{\partial H_{ij}}$, and $\frac{\partial\hat{\boldsymbol{\epsilon}}}{\partial H_{ij}}$ are sensitivity matrices of dimensions $(m\times 1)$,
$(m\times 1)$, $(p\times 1)$, and $(m\times 1)$ respectively. Units of $\frac{\partial\hat{\mathbf{s}}_B}{\partial H_{ij}}$ and $\frac{\partial\hat{\mathbf{s}}_G}{\partial H_{ij}}$ are the same as their kronecker product counterparts.

## 2.2.2 LSA with respect to error covariance matrices

In order to compute the local sensitivities of $\hat{s}$ with respect to $\mathbf{Q}$ and $\mathbf{R}$, consider that they are parametrized as $\mathbf{Q}(\boldsymbol{\theta_Q})$ and $\mathbf{R}(\boldsymbol{\theta_R})$ where $\boldsymbol{\theta_Q}$ and $\boldsymbol{\theta_R}$ are the parameter vectors. The differentiation with respect to error covariance parameters in $\mathbf{Q}$ and $\mathbf{R}$ can be accomplished from Eq. (29) through (32) where the subscript $i$ indicates the $i^{\text{th}}$ covariance parameter for which differentiation is being performed.

$$\frac{\partial \hat{\mathbf{s}}_B}{\partial \theta_{Q_i}} = (\mathbf{I} - \mathbf{CH}) \frac{\partial \mathbf{Q}}{\partial \theta_{Q_i}} \mathbf{H}^t \boldsymbol{\Psi}^{-1}(\mathbf{z} - \mathbf{Hs}_{\text{prior}}) \tag{29}$$

$$\frac{\partial \hat{\mathbf{s}}_G}{\partial \theta_{Q_i}} = \left(-\mathbf{X}\boldsymbol{\Omega}^{-1}\mathbf{A}^T\boldsymbol{\Psi}^{-1}\mathbf{H} + \mathbf{I} - \mathbf{QH}^T\boldsymbol{\Psi}^{-1}\mathbf{H} + \mathbf{QH}^T\boldsymbol{\Psi}^{-1}\mathbf{A}\boldsymbol{\Omega}^{-1}\mathbf{A}^T\boldsymbol{\Psi}^{-1}\mathbf{H}\right) \frac{\partial \mathbf{Q}}{\partial \theta_{Q_i}} \mathbf{H}^T\boldsymbol{\Psi}^{-1}(\mathbf{z} - \mathbf{A}\boldsymbol{\Omega}^{-1}\mathbf{A}^T\boldsymbol{\Psi}^{-1}\mathbf{z}) \tag{30}$$

$$\frac{\partial \hat{\mathbf{s}}_B}{\partial \theta_{R_i}} = -\mathbf{C} \frac{\partial \mathbf{R}}{\partial \theta_{R_i}} \boldsymbol{\Psi}^{-1}(\mathbf{z} - \mathbf{Hs}_{\text{prior}}) \tag{31}$$

$$\frac{\partial \hat{\mathbf{s}}_G}{\partial \theta_{R_i}} = (-\mathbf{X}\boldsymbol{\Omega}^{-1}\mathbf{A}^T - \mathbf{B} + \mathbf{CA}\boldsymbol{\Omega}^{-1}\mathbf{A}^T)\boldsymbol{\Psi}^{-1} \frac{\partial \mathbf{R}}{\partial \theta_{R_i}} \boldsymbol{\Psi}^{-1}(\mathbf{z} - \mathbf{A}\boldsymbol{\Omega}^{-1}\mathbf{A}^T\boldsymbol{\Psi}^{-1}\mathbf{z}) \tag{32}$$

All the quantities $\frac{\partial \hat{\mathbf{s}}_B}{\partial \theta_{Q_i}}$, $\frac{\partial \hat{\mathbf{s}}_G}{\partial \theta_{Q_i}}$, $\frac{\partial \hat{\mathbf{s}}_B}{\partial \theta_{R_i}}$, and $\frac{\partial \hat{\mathbf{s}}_G}{\partial \theta_{R_i}}$ are sensitivity matrices of dimension $(m \times 1)$ and the units of the entries of $\frac{\partial \hat{\mathbf{s}}}{\partial \theta_{Q_i}}$ and $\frac{\partial \hat{\mathbf{s}}}{\partial \theta_{R_i}}$ are of the form $(\mu\text{moles}^{-1}\text{m}^2\text{sec}^{-1})(\text{unit of } \theta_{Q_i} \text{ or } \theta_{R_i})^{-1}$. It is also possible to find $\frac{\partial \hat{\mathbf{s}}}{\partial \mathbf{Q}}$ and $\frac{\partial \hat{\mathbf{s}}}{\partial \mathbf{R}}$ directly as shown in Eq. (33) through (36).

$$\frac{\partial \hat{\mathbf{s}}_B}{\partial \mathbf{Q}} = \mathbf{H}^t \boldsymbol{\Psi}^{-1}(\mathbf{z} - \mathbf{Hs}_{\text{prior}}) \otimes \left(\mathbf{I} - \mathbf{H}^t\boldsymbol{\Psi}^{-1}\mathbf{B}^t\right) \tag{33}$$

$$\frac{\partial \hat{\mathbf{s}}_G}{\partial \mathbf{Q}} = \left(\mathbf{G}_z - \mathbf{z}^t\right) \boldsymbol{\Psi}^{-1}\mathbf{H} \otimes \left(\left(\mathbf{B} - \mathbf{MA}^t + \mathbf{L}^t\mathbf{A}^t\right) \boldsymbol{\Psi}^{-1}\mathbf{H} - \mathbf{I}\right) \tag{34}$$

$$\frac{\partial \hat{\mathbf{s}}_B}{\partial \mathbf{R}} = \boldsymbol{\Psi}^{-1}(\mathbf{z} - \mathbf{Hs}_{\text{prior}}) \otimes \boldsymbol{\Psi}^{-1}\mathbf{HQ} \tag{35}$$

$$\frac{\partial \hat{\mathbf{s}}_G}{\partial \mathbf{R}} = \left(\mathbf{G}_z - \mathbf{z}^t\right) \boldsymbol{\Psi}^{-1} \otimes \left(\mathbf{B} - \mathbf{MA}^t + \mathbf{L}^t\mathbf{A}^t\right) \boldsymbol{\Psi}^{-1} \tag{36}$$

First two quantities $\frac{\partial \hat{\mathbf{s}}_B}{\partial \mathbf{Q}}$ and $\frac{\partial \hat{\mathbf{s}}_G}{\partial \mathbf{Q}}$ are sensitivity matrices of dimension $(m \times m^2)$. The second set of quantities $\frac{\partial \hat{\mathbf{s}}_B}{\partial \mathbf{R}}$ and $\frac{\partial \hat{\mathbf{s}}_G}{\partial \mathbf{R}}$ are sensitivity matrices of dimension $(m \times n^2)$. Equations (33) through (36) are useful when $\mathbf{Q}$ and $\mathbf{R}$ are fully or partially non-parametric. However, dimensions of these matrices can be quite large and users needs to be careful in realizing the full matrix.

## 2.3 Global sensitivity analysis (GSA): a variance-based approach

GSA is a process of apportioning the uncertainty in output to the uncertainty in the input parameters. The term "global" stems from accounting for the effect of all input parameters simultaneously. This is different from LSA, where the impact of a slight change in each parameter on the functional output is considered separately while keeping all other parameters constant. Although quite significant, detailed GSA is challenging as it requires knowledge of the probabilistic variations of all

possible combinations (also known as covariance) of the input parameters,which in most situations is unavailable. However,
sometimes it might be possible to know the approximate joint variation of a small subset of the input parameters (e.g. the
covariance between $\mathbf{Q}$ and $\mathbf{R}$ parameters). Besides the variance-based method, derivative-based global sensitivity measures
or the active-subspace technique (see Appendix A for discussion) can also be used to conduct GSA. However, this work uses
the variance-based method as it does not require sampling and can leverage previously computed partial derivatives. It uses
a first-order Taylor's approximation of parameter estimates to compute global sensitivities. This technique has been used in
many research works, including environmental modeling (e.g., Hamby, 1994) and life cycle assessment (Groen et al., 2017;
Heijungs, 1996), among others.
Broadly, we can consider $\hat{\mathbf{s}}$ as a function of the covariates $\mathbf{Q}, \mathbf{R}, \mathbf{H}, \mathbf{X}$(or $\mathbf{s}_{\mathrm{prior}}$), and $\mathbf{z}$ i.e. $\hat{\mathbf{s}} = \mathbf{f}(\mathbf{Q}, \mathbf{R}, \mathbf{H}, \mathbf{X}$(or $\mathbf{s}_{\mathrm{prior}}), \mathbf{z})$.
We can then compute how uncertainties of the individual components of $\mathbf{f}$ are accounted for in the overall uncertainty of $\hat{\mathbf{s}}$ by
applying multivariate Taylor series expansion of $\hat{\mathbf{s}}$ about its mean. Approximation up to first-order polynomial of the Taylor
series expansion leads to the equation:
$\mathrm{Var}\left(\hat{\boldsymbol{s}}\right) = \left(\dfrac{\partial \hat{\mathbf{s}}}{\partial \boldsymbol{\theta}}^{t} \mathbf{W}_{\boldsymbol{\theta}} \dfrac{\partial \hat{\mathbf{s}}}{\partial \boldsymbol{\theta}}\right)_{\boldsymbol{\theta} = \hat{\boldsymbol{\theta}}} + \mathrm{Error},$
where $\boldsymbol{\theta} = (\boldsymbol{\theta}_{Q}, \boldsymbol{\theta}_{R}, \boldsymbol{\theta}_{H}, \boldsymbol{\theta}_{\mathbf{X}}$(or $\mathbf{s}_{\mathrm{prior}}), \boldsymbol{\theta}_{\boldsymbol{z}})$ is the vector of parameters and $\mathbf{W} = \mathrm{Var}(\boldsymbol{\theta})$ is the covariance matrix of the param-
eters.
It is challenging to estimate covariance quantities such as the cross-covariance between $\boldsymbol{\theta}_{R}$ and $\boldsymbol{\theta}_{H}$ or between $\boldsymbol{\theta}_{H}$, and $\boldsymbol{\theta}_{Q}$
to get the best possible estimate of the total uncertainty of $\hat{\mathbf{s}}$. Assuming no cross-covariance between $\mathbf{Q}$ and $\mathbf{R}$ and ignoring
other parameters not related to the variance parameters, the diagonal of the variance of the posterior fluxes can be approximated
as:
$\mathrm{Var}(\hat{s}_i) = \sum_{j=1}^{L} \left(\dfrac{\partial \hat{\boldsymbol{s}}}{\partial \theta_{Q_j}}\right)_i^2 \mathrm{Var}\left(\theta_{Q_j}\right) + \sum_{k=1}^{M} \left(\dfrac{\partial \hat{\boldsymbol{s}}}{\partial \theta_{R_k}}\right)_i^2 \mathrm{Var}\left(\theta_{R_k}\right)\Bigg|_{\boldsymbol{\theta} = \hat{\boldsymbol{\theta}}},$  (37)
where the subscript $i$ on the right-hand side of Eq. (37) refers to the $i^{\mathrm{th}}$ entry of the derivative vector, which is a scalar and
parameters $\theta_{Q_j}$ and $\theta_{R_k}$ refer to the $j^{\mathrm{th}}$ and $k^{\mathrm{th}}$ parameters of the sets $\boldsymbol{\theta}_Q$ and $\boldsymbol{\theta}_R$ respectively. From Eq. (37), we can see
how uncertainty in the flux estimate is apportioned between variance of $\boldsymbol{\theta}_Q$ and $\boldsymbol{\theta}_R$. No normalization is necessary in such a
framework as, variance components on the right hand side of Eq. (37) are naturally weighted, resulting in the same units of
measurement. Once the two parts of $V_{\hat{s}_i}$ (i.e. Eq. (37)) are computed, they can also be summed over the solution space (e.g.
number of gridcells $\times$ number of periods) of $\hat{\mathbf{s}}$ and ranked to find the relative importance of the parameters.
Even after simplification, implementation of Eq. (37) is complex as it requires knowledge of the uncertainties associated
with the parameters of $\mathbf{Q}$ and $\mathbf{R}$ that are generally not known. We do not further discuss GSA in the context of the case study
presented in this work, but we have shown its application with respect to $\mathbf{Q}$ and $\mathbf{R}$ in the MATLAB Livescript.

Besides the variance-based method, there are many different approaches for performing GSA, as described in Appendix. A.
However, they are either computationally expensive or assume independence of the input parameters, which is not the case
in atmospheric inverse problems. We do not pursue other approaches for quantifying GSA associated with $\mathbf{Q}$ and $\mathbf{R}$ as they
would lead to similar results and would not add anything substantial to the contributions of this study.

## 306   2.4   Ranking importance of covariates, covariance parameters, and observations from LSA

In atmospheric inverse modeling, we encounter two situations while ranking the importance of parameters. These are ranking
of parameters when they have the same or different units. The situation of ranking parameters with the same units arises when
we want to study the influence of a group of parameters, like observations with the same units. Comparatively, the ranking of
parameters with different units occurs when we want to explore the impact of groups of parameters with dissimilar units of
measurements, like observations in $\mathbf{z}$ in comparison to the variance of observations in $\mathbf{R}$. Both these situations can be accounted
for in GSA described in Sec. 2.3. However, GSA in most scenarios in atmospheric inverse modeling cannot be performed due
to the reasons mentioned earlier. Therefore, in this work, we adopt a regression-based approach to rank the importance of
parameters. The proposed approach utilizes output from LSA, accounts for multicollinearity, and results in importance scores
that are bounded between 0 to 1. We define the regression model for ranking as:
$\hat{\mathbf{s}} = \mathbf{E}\boldsymbol{\gamma} + \boldsymbol{\xi},$                  (38)
where $\hat{\mathbf{s}}$ are fluxes obtained from an inversion, and $\mathbf{E}$ is an $(m \times \text{number of derivatives})$ matrix of the previously estimated
sensitivities. The vector of unknown coefficients $\boldsymbol{\gamma}$ is of dimension (number of derivatives $\times$ 1), and $\boldsymbol{\xi}$ is an $(m \times 1)$ vector of
unobserved errors associated with the regression model. To exemplify, $\mathbf{E}$ in Eq. (38) can be arranged as:
$\mathbf{E} = \begin{bmatrix} \dfrac{\partial \hat{\mathbf{s}}}{\partial \mathbf{z}} & \dfrac{\partial \hat{\mathbf{s}}}{\partial \mathbf{Q}} & \dfrac{\partial \hat{\mathbf{s}}}{\partial \mathbf{R}} & \cdot & \cdot \end{bmatrix}$          (39)
In a regression-based approach, as described in Eq. (38), multicollinearity between independent variables in $\mathbf{E}$ can pose a
problem for determining the importance of independent variables in influencing $\boldsymbol{\Gamma}$. To avoid this problem, we compute relative
importance weights by using the method outlined in Johnson, 2000. These weights are obtained by first deriving uncorrelated
orthogonal counterparts of the covariates in $\mathbf{E}$ and then regressing $\hat{\mathbf{s}}$, on $\mathbf{E}$ to get importance weights for each covariate. The
coefficient of determination then standardizes the weights, i.e., $R^2$ such that they range between $0$ to $1$ with the aggregated sum
of 1. Implementation of this method is included in the Livescript submitted with this manuscript.

Note Least Absolute Shrinkage and Selection Operator (LASSO) or Principal Component Analysis (PCA) can also rank
parameters under multicollinearity. However, both these methods result in unbounded weights. Furthermore, "inference after
selection" is ambiguous for LASSO coefficients (see Berk et al., 2013 or chapter 6 of Hastie et al., 2015 for details). Conse-
quently, interpreting the LASSO coefficients as ranks may not be the best approach.

The regression-based approach described above can rank parameters with the same and different units of measurement.
However, an additional normalization step is required to get the overall rank of the parameters with varying units of measure,
like in $\mathbf{z}$, $\mathbf{Q}$, and $\mathbf{R}$. To perform this normalization, first, each column in every sensitivity matrix (e.g. $\frac{\partial \hat{\mathbf{s}}}{\partial \mathbf{z}}$, $\frac{\partial \hat{\mathbf{s}}}{\partial \mathbf{Q}}$, and so forth)
that is to be ranked is normalized (min-max normalization; see Vafaei et al., 2020) between $0$ to $1$. After which, all columns
for a sensitivity matrix are summed and renormalized to vary between $0$ to $1$, resulting in one column representing a sensitivity
matrix for a particular group. We denote this by the subscript "grouped" (e.g. $\frac{\partial \hat{\mathbf{s}}}{\partial \mathbf{z}}_{\text{grouped}}$) in latter sections.

Once the normalized sensitivity vectors are obtained for each group, the regression methodology as described above can be
used to rank the importance of each group. The ranking methodology proposed above does not account for the non-linear rela-
tionship between estimates of the fluxes and the derivatives. If this is a concern, then the strength of the non-linear relationship
among the derivative vectors can be first obtained by computing distance correlation between fluxes and the local derivatives
of the parameters. If necessary, variable transformation techniques such as Box-Cox transformation (see Sakia, 1992) can be
employed before adopting the regression methodology described above.

Note that in most batch inversion methods, DOFS is used to assess the information content provided by observations.
DOFS $= 0$ in these inversions implies that no informational gain happened. In this case, the estimated flux reverts to prior. In
Eq. (38), this means that the $\gamma$ coefficient that corresponds to $\mathbf{Q}$ would have the most significant impact. Likewise if DOFS is
large, then the $\gamma$ coefficients for $\mathbf{z}$ and $\mathbf{R}$ would be larger (and likely correlated). We show this correspondence in Sec. 3.

Finally, all diagnostic methods applied in the context of any regression-based model can be used to understand the rela-
tionship between dependent and independent variables; however, what covariates to include in $\mathbf{E}$ depends on the specific case
study under consideration.

## 3   Results

To demonstrate the applicability of our methods, we utilize data from our published work on $CH_4$ fluxes in the Los Angeles
megacity (see Yadav et al., 2019). In this previous work, fluxes were estimated for South Coast Air Basin (SoCAB) region (see
Fig. 3) at $0.03°$ spatial (1826 grid-cells) and 4-day temporal resolution from the Jan 27, 2015 through Dec 24, 2016. However,
in the current work, we utilize input data from Oct 23, 2015, through Oct 31, 2015, which is a single inversion period, to
contextualize the applicability of our methods. This period overlaps with the beginning of the well-studied Aliso Canyon gas
leak (Conley et al., 2016). As in previous work, $\mathbf{R}$ and $\mathbf{Q}$ are assumed to be diagonal with separate parameter for each site in $\mathbf{R}$
and a single parameter that governs the scaling of errors in $\mathbf{Q}$. Similarly, $\mathbf{X}$ is a column vector consisting of the prior estimates
of CH$_4$ fluxes.

For each observation included in the case study, a forward operator was obtained by using Weather Research Forecasting-
Stochastic Time Inverted Lagrangian Model (see Yadav et al., 2019). These forward operators are used to demonstrate the
application of the methodology for building IAOMI and JSD-based correlation matrices in the MATLAB Livescript. They are
also used with measurements and prior information to estimate the fluxes and perform LSA.
**3.1 STAD from the forward operators**
In this work, we identify STAD for the 4-day period for which the inversion was performed. The spatial domain of the study
over this period is uniquely disaggregated by STAD, as shown in Fig. 3. The STAD for different sites is mostly spatially
contiguous. Still, for some monitoring sites, we found isolated grid cells that were not within the adjacent zones. We manually
combined these with STAD for the nearest site to create a spatially continuous map, as shown in Fig. 3. The discontinuous
version of the STAD shown in Fig. 3 is included in the Livescript. The discontinuities in the STAD result mainly from an
unequal number of observations across sites and indicate that aggregation over a more extended period is required to identify
a noise-free STAD. We do not investigate the period of this aggregation as this is beyond the scope of this work.

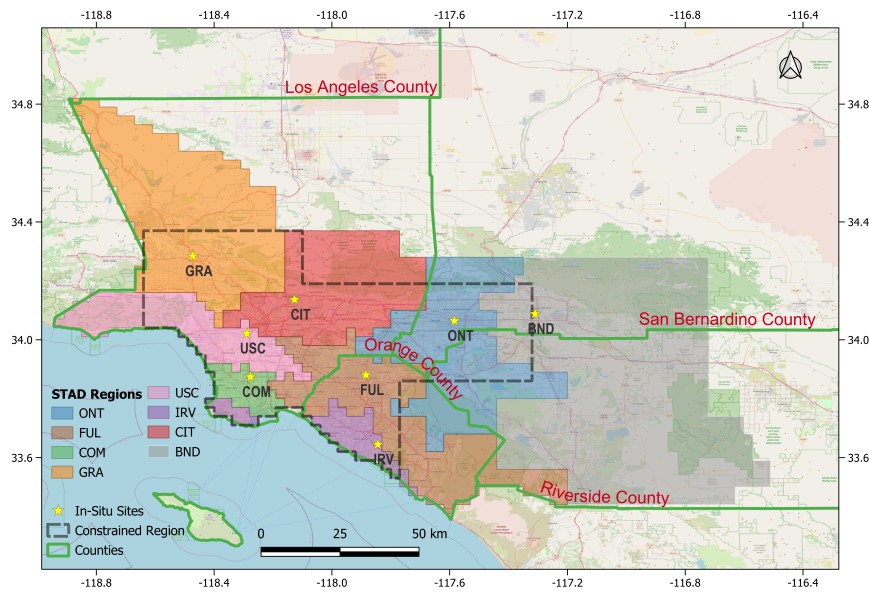

**Figure 3.** Study area with county boundaries, measurement locations and the Spatio-temporal Area of Dominance of measurement locations. The black dotted line shows the area constrained by observations, as shown in Yadav et al., 2019.

Overall, STAD for each site indicates spatial regions of fluxes over a period that contribute most to the observational signal
observed at a site allowing us to associate the change in fluxes to the specific area in the basin where reductions or increases
in emissions are likely to have occurred. Some information in the observational signal is shared between observations from
different sites. This shared information (though not shown) can be computed as part of STAD and forms part of overall basin-
scale estimates of fluxes that combines measurements from all sites. Note that STAD does not represent the network's coverage,
i.e., regions of emissions constrained by observations. These regions are shorter than STAD (see the grey outline in Fig. 3).
They are obtained before performing an inversion by identifying areas of continuous spatiotemporal coverage as provided by
atmospheric transport (Fig. 4) or by assessing the model resolution after performing an inversion (for an explanation, see Yadav
et al., 2019).

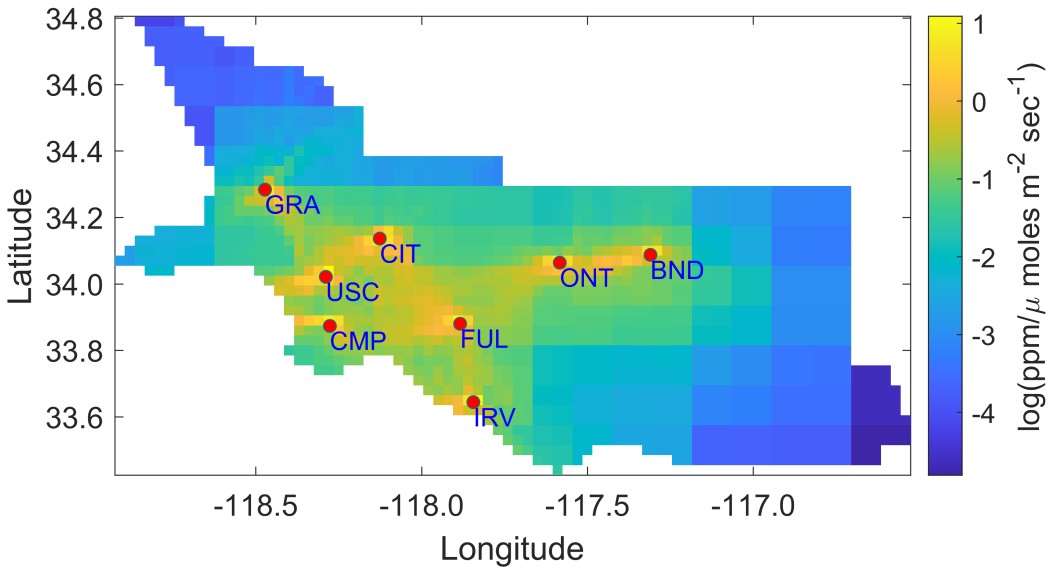

**Figure 4.** Heatmap of the aggregated forward operators for the case study period.

## 3.2 Sensitivity analysis

One of the main goals of the sensitivity analysis after performing an inversion is to identify the observations that had the most
influence on the flux estimates. Other than observations, it is also essential to explore the importance of different inputs to an
inversion, like variance parameters in $\mathbf{R}$. We describe the process of performing this analysis within the context of the case
study mentioned in Sec. 3, which discusses the relative importance of the input quantities in influencing $\hat{s}$, by utilizing local
sensitivities.

| Site | Importance Score | Rank |
|------|-----------------|------|
| GRA | 0.26 | 1 |
| ONT | 0.24 | 2 |
| COM | 0.13 | 3 |
| IRV | 0.11 | 4 |
| BND | 0.10 | 5 |
| CIT | 0.07 | 6 |
| FUL | 0.07 | 7 |
| USC | 0.06 | 8 |

**Table 1.** The importance scores and ranking of 8 sites based on the sensitivity of the estimated fluxes ($\hat{s}$) to observations ($z$).

### 3.2.1 Comparison and ranking of the observations

Importance of individual measurements in influencing $\hat{s}$, can be easily computed through the relative importance methodology described in section 2.4. Although all entries of $\frac{\partial \hat{s}}{\partial z}$ are in same units of measurement, direct ranking of observations or sites without employing the relative importance technique can lead to misleading results, which happens due to the presence of large negative and positive values in $\frac{\partial \hat{s}}{\partial z}$ that are governed by the overall spatiotemporal spread, the intensity of forward operators, and high enhancements.

For the case study in this work, we find that observations collected at the GRA site that is located nearest to the source of the Aliso Canyon gas leak are most influential in governing $\hat{s}$, as shown by site-based rankings in Table 1. These rankings primarily show the importance of observations from a site in influencing the estimated fluxes for the period in consideration and are obtained by summing the weights for each observation by employing the relative importance methodology.

Outliers have a significant impact on these rankings. The high weight associated with even one observation from a site can make that site more important compared to other sites. For example, if we remove the observation with the highest weight from each site, ONT is the most important site, followed by GRA, CMP, IRV, CIT, FUL, BND, and USC. As part of sensitivity analysis, examining the influence of the observations associated with high weights is crucial because they are likely to have an enormous impact on the flux estimates. Site level importance should be judged not only by examining the aggregated ranking as presented in Table 1 but also by looking at the distribution of weights shown through the boxplot in the Livescript associated with section 3.2. A site with evenly distributed weights is more important than one whose importance is just due to the presence of a few observations with high weights.

The ranking of each observation in influencing the estimates of fluxes can be obtained by examining the weights of the column vectors of $\frac{\partial \hat{s}}{\partial z}$, and is provided in the Livescript. To exemplify, this ranking of weights showed that observation from

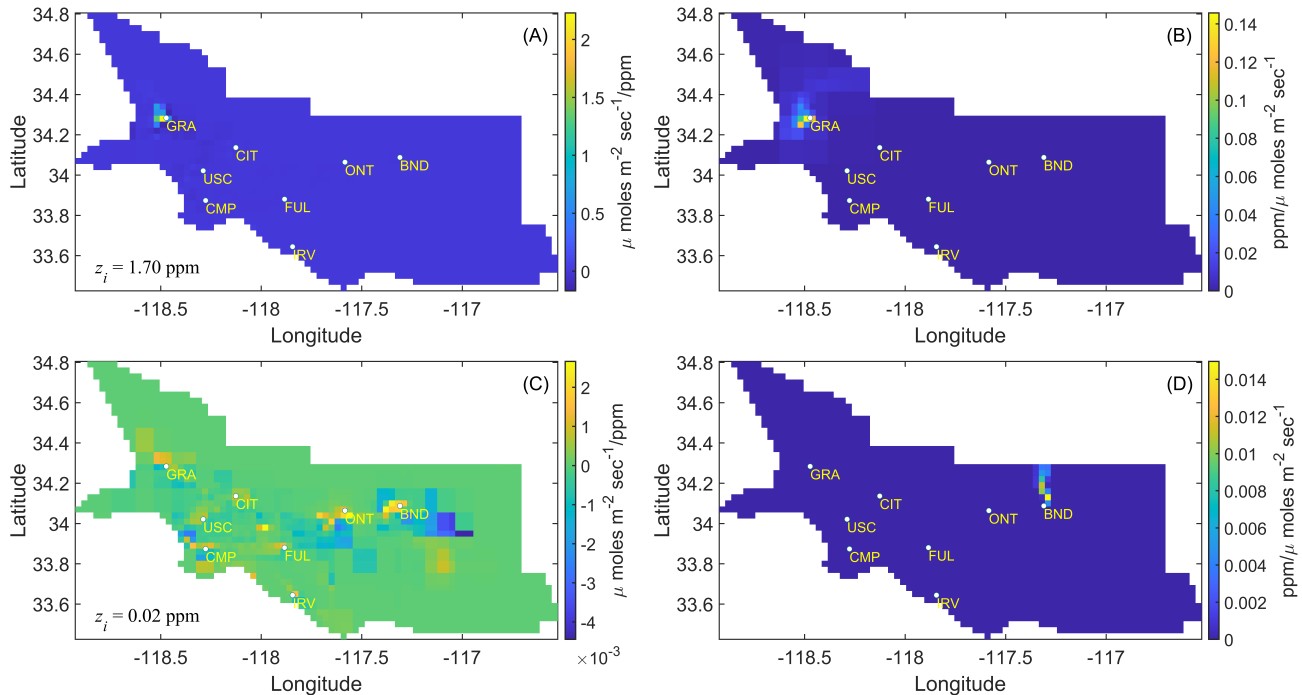

**Figure 5.** The sensitivities ($\frac{\partial \hat{s}}{\partial z_i}$) and forward operators of the most and least important observations are shown here. Subplots A and C depict the sensitivity of $\hat{s}$ with respect to the most (A) and least (C) important observation, respectively, during the case study period. The $CH_4$ enhancement corresponding to these observations is shown in the bottom left corner of the subplots and denoted by the symbol $z_i$. The right subplots, B and D, display the forward operators associated with the sensitivities shown in subplots A and C, respectively.

414 the GRA site with the enhancement of 1.7 ppm was most important, whereas an observation from the BND site with an

415 enhancement of 0.02 ppm was found to be least important in influencing $\hat{s}$. Note this is not an observation with the lowest

416 enhancement but with the least influence (Fig. 5).

417 **3.2.2 Relative importance of $Q, R, X, \beta,$ and $z$**

418 After the two-step normalization of $\frac{\partial \hat{s}}{\partial z}$, $\frac{\partial \hat{s}}{\partial X}$, $\frac{\partial \hat{s}}{\partial H}$, $\frac{\partial \hat{s}}{\partial \beta}$, $\frac{\partial \hat{s}}{\partial Q}$, and $\frac{\partial \hat{s}}{\partial R}$ as described in section 2.4 the spatial plots of all these

419 grouped quantities that we call as $\frac{\partial \hat{s}}{\partial z}_{\text{grouped}}$, $\frac{\partial \hat{s}}{\partial X}_{\text{grouped}}$, $\frac{\partial \hat{s}}{\partial H}_{\text{grouped}}$, $\frac{\partial \hat{s}}{\partial \beta}_{\text{grouped}}$, $\frac{\partial \hat{s}}{\partial Q}_{\text{grouped}}$, and $\frac{\partial \hat{s}}{\partial R}_{\text{grouped}}$ can be created to explore

420 the regions of the low and high weights (see Fig. 6) at the grid scale.

421

422  Some of these quantities are correlated and should be seen in conjunction. For example, $R$ describes errors in $z$, among

423 other errors, and implies that $\frac{\partial \hat{s}}{\partial R}_{\text{grouped}}$ and $\frac{\partial \hat{s}}{\partial z}_{\text{grouped}}$ should be evaluated together to understand their importance in influencing

424 flux estimates. Similarly $Q$ describes errors in $s - X\beta$ implying that $\frac{\partial \hat{s}}{\partial Q}_{\text{grouped}}$ and $\frac{\partial \hat{s}}{\partial X}_{\text{grouped}}$, should be assessed together to

425 understand their importance in influencing flux estimates. A larger value of $\frac{\partial \hat{s}}{\partial z}_{\text{grouped}} + \frac{\partial \hat{s}}{\partial R}_{\text{grouped}}$ is likely to be found around

in-situ sites due to increased model resolution. However, if around these locations $\frac{\partial \hat{\mathbf{s}}}{\partial \mathbf{R}}_{\text{grouped}}$ is larger in comparison of $\frac{\partial \hat{\mathbf{s}}}{\partial \mathbf{z}}_{\text{grouped}}$
then it suggests that errors in $\mathbf{R}$ should be adjusted and therefore observations should be more important in governing the flux
estimates around in-situ sites. In this case study, this is due to the large variability in the enhancement caused by the Aliso
Canyon leak and the presence of large point sources near in-situ sites. Overall, for the exact location, a larger $\frac{\partial \hat{\mathbf{s}}}{\partial \mathbf{z}}_{\text{grouped}_i}$ should
be accompanied by a lower $\frac{\partial \hat{\mathbf{s}}}{\partial \mathbf{R}}_{\text{grouped}_i}$, as confirmed by the correlation subplots A and B of Fig. 7.
The increased model resolution also results in lower importance of $\frac{\partial \hat{\mathbf{s}}}{\partial \mathbf{X}}_{\text{grouped}}$ and $\frac{\partial \hat{\mathbf{s}}}{\partial \mathbf{Q}}_{\text{grouped}}$, around sites. However, areas
unconstrained by observations are likely to have larger $\frac{\partial \hat{\mathbf{s}}}{\partial \mathbf{X}}_{\text{grouped}} + \frac{\partial \hat{\mathbf{s}}}{\partial \mathbf{Q}}_{\text{grouped}}$ as seen in Fig. 6 for $\frac{\partial \hat{\mathbf{s}}}{\partial \mathbf{X}}_{\text{grouped}}$ and $\frac{\partial \hat{\mathbf{s}}}{\partial \mathbf{Q}}_{\text{grouped}}$,
quantities. If in locations constrained by observations, $\frac{\partial \hat{\mathbf{s}}}{\partial \mathbf{Q}}_{\text{grouped}_i}$ is larger in comparison to $\frac{\partial \hat{\mathbf{s}}}{\partial \mathbf{X}}_{\text{grouped}_i}$ then $\mathbf{X}$ in these locations
is incorrect and needs adjustment. Likewise, in the case of $\frac{\partial \hat{\mathbf{s}}}{\partial \mathbf{R}}_{\text{grouped}}$ a larger $\frac{\partial \hat{\mathbf{s}}}{\partial \mathbf{X}}_{\text{grouped}_i}$ is generally accompanied by lower
$\frac{\partial \hat{\mathbf{s}}}{\partial \mathbf{z}}_{\text{grouped}_i}$ and vice versa, which is also visible in the correlation subplots C and D in Fig. 7. Quantity $\frac{\partial \hat{\mathbf{s}}}{\partial \boldsymbol{\beta}}_{\text{grouped}}$ provides
information about the grid-cells that are determining the value of $\hat{\beta}$ and in this case study as expected this is around Aliso
Canyon leak whose $X_i$ is being adjusted due to the larger flux from that region. This can also be seen in subplot E in Fig. 7
where it is positively correlated with $\hat{\mathbf{s}}$.

## 4   Discussion

This study lays out techniques to assess the quality of the inferred estimates of fluxes. Sensitivity analysis is an important
diagnostic tool to understand the impact of the choices made with respect to inputs on the estimated fluxes. However, it is not
a recipe for selecting the proper forms of $\mathbf{X}$ or the structure of $\mathbf{Q}$ or $\mathbf{R}$ before performing an inversion. Other tools or methods
such as Bayesian Information Criterion, Variance Inflation Factor should be used to perform this task.
The case study in this work is designed only to demonstrate the methodologies described in Sec. 2. We do not impose non-
negativity constraints to obtain positive CH$_4$ fluxes as was done in the original 2019 study (Yadav et al., 2019). This is done
because posterior likelihood changes its functional form under non-negativity constraints that invalidate the analytical forms
of sensitivity equations presented in this work. Thus, some CH$_4$ fluxes obtained in this study have negative values as can be
seen in the map of $\hat{\mathbf{s}}$ in the MATLAB Livescript. Even in these situations assessing sensitivity through an inversion without
the imposition of non-negativity is helpful as it provides insights into the role of $\mathbf{z}$, $\mathbf{R}$, $\mathbf{Q}$, and $\mathbf{X}$ in governing estimates of
non-negative $\hat{\mathbf{s}}$.
Like $\mathbf{z}$, the importance of $\mathbf{Q}$ and $\mathbf{R}$ parameters can be directly obtained when all parameters have the same units of measure-
ment as in the case study presented in this study. However, this is not guaranteed as $\mathbf{R}$ can be a function of variance parameters
and spatiotemporal correlation lengths expressed in the distance units in space and time. Furthermore, a nonstationary error
covariance $\mathbf{R}$ can have parameters that have even more complicated units. This situation is not limited to $\mathbf{R}$ but also applies to
the prior error covariance $\mathbf{Q}$ and $\mathbf{X}$. Under these conditions, comparing the sensitivity matrices is only possible after normal-
ization. Therefore, we recommend using a multiple linear regression-based relative importance method to rank these quantities

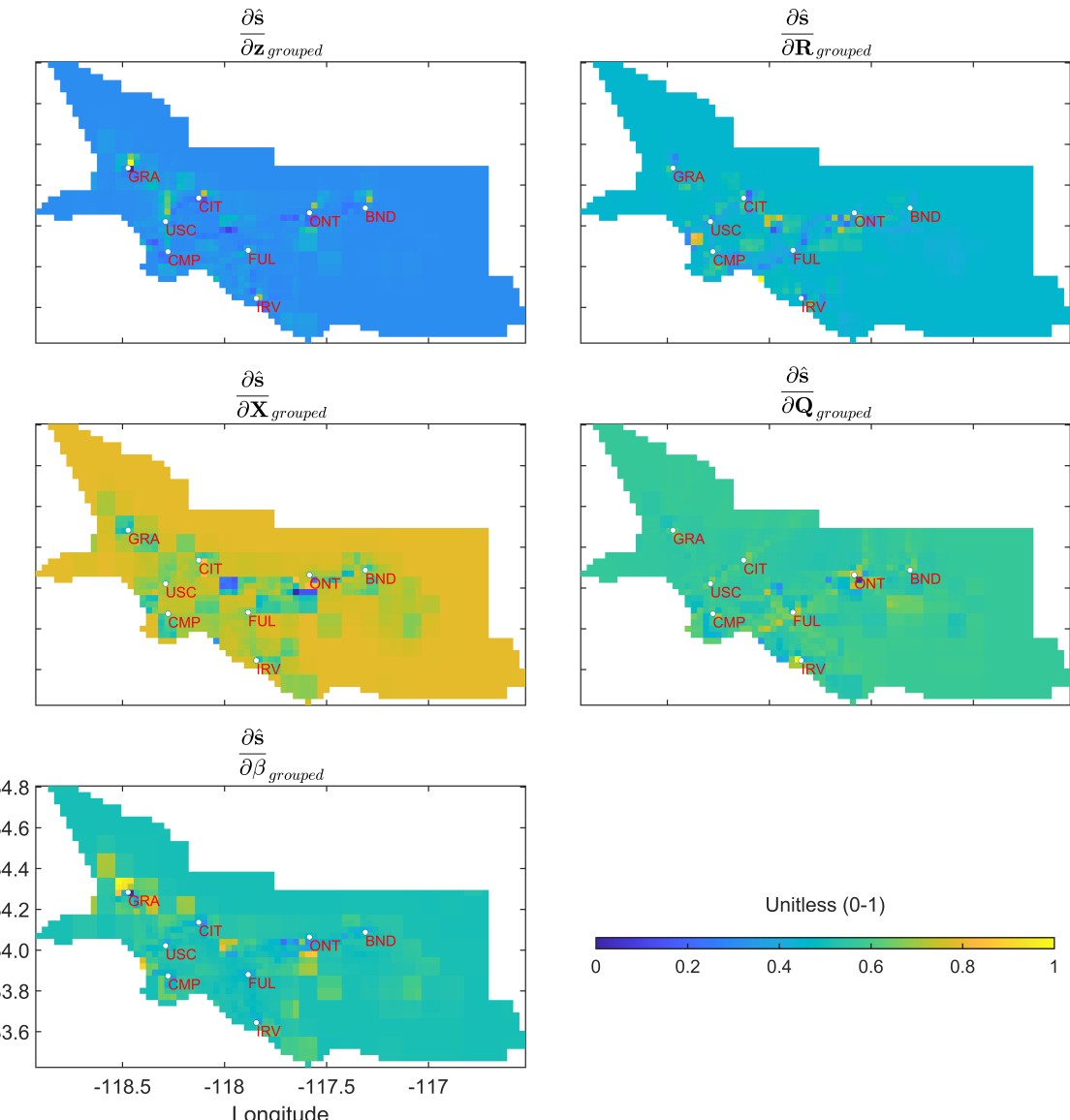

**Figure 6.** Grouped local sensitivities of the estimated fluxes (ŝ) with respect to **z**, **R**, **X**, **Q**, and $\boldsymbol{\beta}$ from top-left to bottom-right respectively. Note, in the case of $\frac{\partial \hat{\mathbf{s}}}{\partial \mathbf{z}}_{\text{grouped}}$, $\frac{\partial \hat{\mathbf{s}}}{\partial \mathbf{R}}_{\text{grouped}}$, and $\frac{\partial \hat{\mathbf{s}}}{\partial \mathbf{X}}_{\text{grouped}}$ two-step normalization is performed to generate subplots associated with these quantities. Derivatives with respect to: (1) observations in **z**, (2) parameters in **R**, and (3) entries in **X** are normalized between 0 and 1 and then after aggregating these for every grid-cell another Min-Max normalization is performed to limit their ranges between 0 and 1. Only single normalization is performed in case of $\frac{\partial \hat{\mathbf{s}}}{\partial \mathbf{Q}}_{\text{grouped}}$ and $\frac{\partial \hat{\mathbf{s}}}{\partial \boldsymbol{\beta}}_{\text{grouped}}$ as they consist of only one parameter.

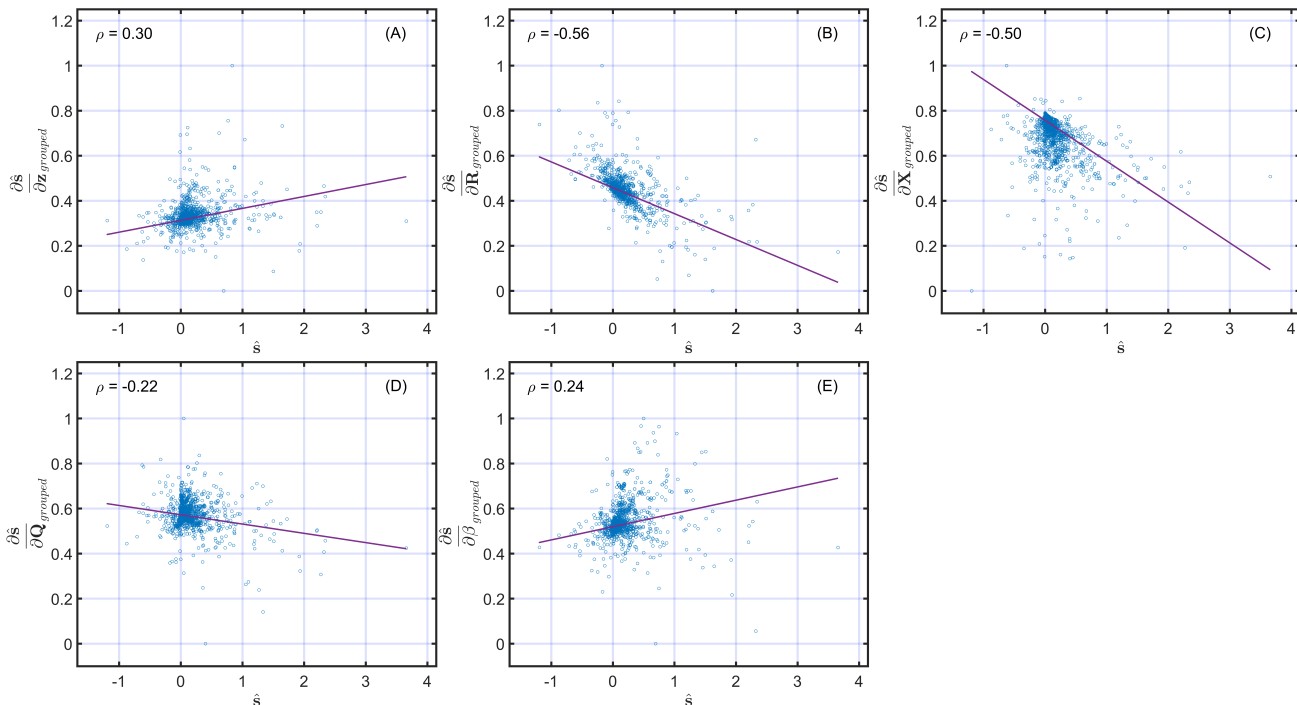

**Figure 7.** Scatterplots of relationships between $\hat{s}$ and $\frac{\partial \hat{s}}{\partial \mathbf{z}}_{\text{grouped}}$, $\frac{\partial \hat{s}}{\partial \mathbf{R}}_{\text{grouped}}$, $\frac{\partial \hat{s}}{\partial \mathbf{X}}_{\text{grouped}}$, $\frac{\partial \hat{s}}{\partial \mathbf{Q}}_{\text{grouped}}$, $\frac{\partial \hat{s}}{\partial \beta}_{\text{grouped}}$. Note as before in Fig. 6 all the derivatives are normalized to limit their range between 0 and 1. The correlation coefficient of the relationships shown in each scatterplot is reported on the top right corner of the subplots. The least square line of best fit is shown in red color in every subplot.

for comparative assessment.

The overall importance of $\frac{\partial \hat{s}}{\partial \mathbf{z}}$ is best explored by performing column-based normalization and then employing the relative
importance method. Additionally, column based normalization can be augmented by row-based normalization to assess and
rank the influence of observations in governing grid-scale estimates of $\hat{s}$. Qualitatively, column and row-based assessment
increase our understanding about the spatiotemporal estimates of $\hat{s}$ˆ, which is especially important when point sources are the
dominant sources of emissions. Moreover, it provides insight into the temporal aggregation error (e.g. Thompson et al., 2011)
as the information encoded in an instantaneous measurement can get lost over the coarser inversion period. This aggregation
error also manifests spatially and is determined by the resolution at which fluxes are obtained. In many situations, these ag-
gregation errors are unavoidable as the choice of the spatiotemporal resolution of inversions is governed by the density of
observations in space and time.

Other than aggregation error, the aggregation of the estimated fluxes also has profound implications as it affects the robust-
ness of the estimated fluxes. It can be proved (see Appendix C1) that aggregation of $\hat{s}$ in space and time from an inversion
conducted at finer resolution leads to reduction in uncertainty. However, even though ratio of observations to the estimated
fluxes increases, the number of fluxes uniquely resolved declines at coarser resolution (see Appendix C2).

The computational cost to calculate analytical partial derivatives is minimal as it is a onetime operation and is bounded
by the computational cost to perform matrix multiplications, which at max is $O(n^3)$. For the case study presented, we can
compute analytical derivatives and rank for approximately $4000$ parameters in few minutes on a laptop. Computing derivatives
by using the Kronecker form of equations (Eq. (18), (21) through (24), and (33) though (36)) is faster for smaller problems.
However, for large problems, the storage costs associated with these equations can become prohibitive. In these situations, we
propose the use of $ij$ form of the equations (Eq. (20), (25) through (28), and (29) though (32)) for assessment. Furthermore,
computational problems can also arise in ranking the inputs if we have numerous derivatives (e.g. greater than 10,000), as the
ranking method used in this work relies on eigenvalue decomposition that has $O(n^3)$ computational complexity. To overcome
this problem, we advise grouping of derivatives to reduce the dimension of the problem.

Finally, the estimation of STAD and the importance of sites can be influenced by data gaps; therefore, it is not advised in
presence of vast differences in the number of observations between sites.
**5  Conclusions**
Our work makes a novel and significant contributions that can improve the understanding of linear atmospheric inverse prob-
lems. It provides (1) a framework for post hoc analysis of the impact of inputs on the estimated fluxes and (2) a way to
understand the correlations in the forward operators or atmospheric transport model. The authors are unaware of any work
where local sensitivities with different units of measurement are compared to rank the importance of inputs in a linear atmo-
spheric inverse model.

Concerning forward operators, we provide mathematical foundations for IAOMI and JSD-based metrics. These two metrics
can be used to construct a nonstationary error covariance for the atmospheric transport component of the model-data mismatch
matrix $\mathbf{R}$. Furthermore, IAOMI-based assessments can be extended to identify STAD from forward operators that can help
in disaggregating regions of influence of the observations over a chosen temporal duration. This assists in understanding the
connection between the sources of fluxes and observations from a particular measurement location.

The IAOMI and JSD-based metrics provide an essential insight into the two critical and only required components for an
inversion: observations and forward operators (e.g., the influence of observation to the sources of fluxes through STAD), which
can be accomplished before conducting an inversion and should be complemented by post hoc LSA, which is necessary for
understanding the behavior of an inverse model. Overall, LSA can answer questions like for which locations and in what order
of precedence was an observation important in influencing the estimated fluxes. This kind of analysis is entirely different from
estimating uncertainty, which tells us the prior uncertainty reduction due to observations.

LSA is not a replacement for statistical tests that check inverse models' underlying assumptions and model specifications.
Neither is it a recipe for selecting inputs to an inverse model. However, as explained above, it has an essential role that can lead
to an improved understanding of an atmospheric inverse model.


© 2022, Jet Propulsion Laboratory, California Institute of Technology
*Code and data availability.* All the code and data utilized in this study are submitted as supplementary material.
**Appendix A: Review of previously employed methods to conduct sensitivity analyses**
Earlier, many methods have been proposed and utilized to perform sensitivity analysis. These can be categorized as global
and local sensitivity analyses. Global sensitivity analysis (GSA) includes Morris's (e.g. Morris, 1991) one step at a time
method (OAT), Polynomial Chaos Expansion (PCE) (e.g. Sudret, 2008), Fourier amplitude sensitivity test (FAST) (e.g. Xu
and Gertner, 2011), Sobol's method (e.g. Sobol, 2001) and Derivative based global sensitivity measures (DGSM) (e.g. Sobol
and Kucherenko, 2010) among others. These existing GSA methods (1) assume independence of parameters (e.g., FAST and
OAT), or (2) computationally expensive (e.g., Sobol's method), or (3) require knowledge of the joint probability distribution of
the parameter space (e.g., DGSM, PCE). Therefore, these traditional methods cannot be directly applied in linear atmospheric
inverse problems, which consists of tens of thousands of non-normal, spatiotemporally correlated parameters (including ob-
servations). Constantine and Diaz, 2017 proposed an active subspace-based GSA that uses a low-dimensional approximation
of the parameter space. But it is still computationally expensive for problems with thousands of parameters (see case study in
Constantine and Diaz, 2017).

Compared to GSA, a local sensitivity method like Bayesian Hyper Differential Sensitivity Analysis (HDSA) (Sunseri et al.,
2020) computes partial derivatives concerning maximum a posteriori probability (MAP) estimates of a quantity of interest.
However, unlike Bayesian HDSA, we do not generate samples from the prior estimate to compute multiple MAP points since
we have limited knowledge of the prior distribution of the spatiotemporally correlated parameters. We derive the functional
form of the local sensitivity equations based on the closed-form MAP solution. Our method is simple and amenable to tens
of thousands of parameters. Note that, like all linear atmospheric inverse problems, one of the critical goals of this work is
to study the importance of thousands of spatiotemporally varying parameters by ranking them, and computation of the local
sensitivities is a means to achieve that goal.

## Appendix B: Jensen-Shannon distance (JSD) for forward operators

The dissimilarity between forward operators can also be measured via entropy (for definition, see MacKay et al., 2003) based distances, which can capture differences between two probability distributions. One such metric is Jensen-Shanon distance (JSD) (Nielsen, 2019), which can be used to compute the distance between two forward operators after normalizing them by their total sum. For a forward operator F this can be given as:

$$P_{F_k} = \frac{F_k}{\sum_k F_k} \tag{B1}$$

where $F_k$ denotes $k^{\text{th}}$ entry of $\mathbf{F}$ resulting in normalized forward operator $P$. We can then use JSD to compute distance between two normalized forward operators from equation B2:

$$JSD(P_{\mathbf{F}}||P_{\mathbf{G}}) = \sqrt{\frac{1}{2}D(P_{\mathbf{F}}||M) + \frac{1}{2}D(P_{\mathbf{G}}||M)} \tag{B2}$$

where $D$ stands for Kulback-Leibler (KL) divergence (see MacKay et al., 2003 for details). KL divergence D of any probability distribution $p$ with respect to another probability distribution $q$ is defined as: $D(p||q) = \sum p\log(p/q)$ and $M$ stands for $\frac{1}{2}(P_{\mathbf{F}} + P_{\mathbf{G}})$. The symbol $||$ is used to indicate that $D(P_{\mathbf{F}}||M)$ and $D(P_{\mathbf{G}}||M)$ are not conditional entropies (see MacKay et al., 2003). JSD is closed and bounded in $[0,1]$ when KL divergence is computed with base 2 logarithm. Intuitively, JSD and $1 - \nu$ (i.e. 1-IAOMI) are comparable since both of them are measures of dissimilarity.

## Appendix C: Uncertainty and model resolution under aggregation

Here we show the proofs of two mathematical statements on the robustness and quality of the estimated fluxes as mentioned in Sec. 4. First, we show why marginal variance of the estimated fluxes (which is the diagonal of covariance matrix of ŝ) decrease when estimated fluxes are post aggregated to a coarser scale or upscaled (A). Second, we show why in such case the model resolution (also termed as, total information resolved by the observations) also decreases (B). Note that, the nomenclature used in the appendix should not be confused with the nomenclature introduced in Sec. 2. The abbreviations and symbols used here are independent of what are used in the Sec. 2.

### C1  Proof of the reduction of marginal variance of ŝ when aggregation is performed

Post inversion aggregation or upscaling of any flux field s is equivalent to pre-multiplication by a weight matrix (in fact, a row stochastic matrix). This can be written as:

$$\tilde{\mathbf{s}} = \mathbf{J}\hat{\mathbf{s}}, \tag{C1}$$

where $\mathbf{J}$ is a row stochastic (i.e. row-sums are all unity) $k \times m$ weight matrix ($k < m$). Variance of $\tilde{s}$ can be written as $\mathbf{J}\Sigma\mathbf{J}^t$
where $\text{var}(\tilde{s}) = \mathbf{J}\text{var}(\hat{s})J^t = \mathbf{J}\Sigma\mathbf{J}^t$. The general structure of $\mathbf{J}$ is as follows:
$$
J = \begin{bmatrix} 0 & j_{12} & j_{13} & \mathbf{0} & \mathbf{0} & \mathbf{0} \\ j_{21} & \mathbf{0} & j_{2r+1} & j_{2r+2} & \mathbf{0} & \mathbf{0} \\ \vdots & \vdots & \ddots & \ddots & \vdots & \vdots \\ \mathbf{0} & \mathbf{0} & \mathbf{0} & j_{km} & \mathbf{0} & \mathbf{0} \end{bmatrix} = \begin{bmatrix} \mathbf{j}_1^t \\ \mathbf{j}_2^t \\ \vdots \\ \mathbf{j}_k^t \end{bmatrix}
\tag{C2}
$$

However, $\mathbf{J}$ is mostly sparse, with non-zero values in only a few places. The rest of the entries are zeros. Essentially, $\mathbf{J}$ can
have any number of non-zero entries in a row that may or may not be consecutive. This is because, although adjacent grids
are averaged on a map, they may not be adjacent upon vectorization. Moreover, the geometry of the map may not be exactly
square or rectangular. Therefore, depending on the aggregation or upscaling factor and geometry, there may or may not be
any neighboring grid for averaging around a particular grid. However, the rows are linearly independent, as nearby grids are
considered only once for averaging. The properties of $\mathbf{J}$ are as follows:

570         1. $\mathbf{J}\mathbf{1} = 1$ or $\mathbf{j}_i^t\mathbf{1} = 1 \quad \forall i = 1, 2, \cdot, \cdot, k$

571         2. $\mathbf{j}_i^t\mathbf{j}_r = 0$ for $i \neq r$

We can rearrange the columns of $\mathbf{J}$ and the rows of $\Sigma$ accordingly without loss of any structure such that non-zero entries
are consecutive for each row of $\mathbf{J}$. Matrix $\mathbf{J}\Sigma\mathbf{J}'$ under column permutation can be written as:
$\mathbf{J}\Sigma\mathbf{J^t} = \mathbf{J}_\pi\Sigma_\pi\mathbf{J}_\pi^t =$
$$
\begin{bmatrix} \mathbf{l}_1^t & 0 & \dots & 0 \\ 0 & \mathbf{l}_2^t & \dots & 0 \\ \vdots & \vdots & \ddots & . \\ 0 & 0 & \dots & \mathbf{l}_k^t \end{bmatrix}^{k \times m} \begin{bmatrix} \Xi_{11} & \Xi_{12} & \dots & \Xi_{1k} \\ \Xi_{21} & \Xi_{22} & \dots & . \\ \vdots & \vdots & \ddots & . \\ \Xi_{k1} & . & \dots & \Xi_{kk} \end{bmatrix}^{m \times m} \begin{bmatrix} \mathbf{l}_1 & 0 & \dots & 0 \\ 0 & \mathbf{l}_2 & \dots & 0 \\ \vdots & \vdots & \ddots & . \\ 0 & 0 & \dots & \mathbf{l}_k \end{bmatrix}^{p \times k}
\tag{C3}
$$

$$
= \begin{bmatrix} \mathbf{l}_1^t\Xi_{11}\mathbf{l}_1 & . & \dots & \mathbf{l}_1^t\Xi_{1k}\mathbf{l}_k \\ . & \mathbf{l}_2^t\Xi_{22}\mathbf{l}_2 & \dots & . \\ \vdots & \vdots & \ddots & . \\ \mathbf{l}_k^t\Xi_{k1}\mathbf{l}_1 & . & \dots & \mathbf{l}_k^t\Xi_{kk}\mathbf{l}_k \end{bmatrix}^{k \times k}
\tag{C4}
$$

where $\mathbf{J}_\pi$ and $\Sigma_\pi$ are the permuted $\mathbf{J}$ and $\Sigma$ respectively. However, for notational clarity, we use $\mathbf{l}$ and $\Xi$ as the sub-vector
and sub-block-matrix of the $\mathbf{J}_\pi$ and $\Sigma_\pi$ respectively. Note that, any $\mathbf{l}_i^t$ is a row-vector of dimension $(1, d_i)$, and $\Xi_{ii}$ is a square
matrix of dimension $(d_i, d_i)$ where $\sum_{i=1}^{k} d_i = m$. Thus, diagonal entry $\mathbf{l}_i^t\Xi_{ii}\mathbf{l}_i$ is a scalar quantity. For any $i^{\text{th}}$ diagonal entry,
the corresponding scalar quantity can be written as $\sum_{jrl} l_{ij}l_{ir}\Xi_{jr}$. By symmetry of $\Xi$, this reduces to
$\mathbf{l}_i^t\Xi_{ii}\mathbf{l}_i = \sum_r l_{ir}^2\Xi_{lr}^2 + 2\sum_{j>r} l_{ij}l_{ir}\Xi_{jr}$
                                                  (C5)

By Cauchy Squartz inequality on $\Xi_{jr}$, this can be written as

$$\sum_r l_{ir}^2 \sigma_{lr}^2 - 2\sum_{j>r} l_{ij}l_{ir}\sigma_{jj}\sigma_{rr} \le \sum_r l_{ir}^2 \sigma_{rr}^2 + 2\sum_{j>r} l_{ij}l_{ij}\sigma_{jr} \le \sum_r l_{ir}^2 \sigma_{rr}^2 + 2\sum_{j>r} l_{ij}l_{ij}\sigma_{jj}\sigma_{rr} \tag{C6}$$

$$\left( l_{ir}\sqrt{\sigma_{ir}} - \sum_{r\ge 2} l_{ir}\sqrt{\sigma_{ir}} \right)^2 \le \sum_r l_{ir}^2 \sigma_{rr}^2 + 2\sum_{j>r} l_{ij}l_{ij}\sigma_{jj}\sigma_{rr} \le \left( \sum_{ir} l_{ir}\sqrt{\sigma_{rr}} \right)^2 \tag{C7}$$

$$\min_r \sigma_{rr}\left( l_{ir} - \sum_{r\ge 2} l_{ir} \right)^2 \le \sum_r l_{ir}^2 \sigma_{rr}^2 + 2\sum_{j>r} l_{ij}l_{ij}\sigma_{jj}\sigma_{rr} \le \max_r \sigma_{rr}\left( \sum_{ir} l_{ir} \right)^2 \tag{C8}$$

This implies (by property 1 of the weight matrix $\mathbf{J}$) that the $i^{\text{th}}$ diagonal entry is bounded by:

$$\min_r \sigma_{rr}\left( l_{ir} - \sum_{r\ge 2} l_{ir} \right)^2 \le \mathbf{J'_i}\mathbf{\Sigma_{ii}}\mathbf{J_i} \le \max_r \sigma_{rr} \le \sum_{r=1}^{d_i} \sigma_{rr} \tag{C9}$$

where $\sum_{r=1}^{d_i}\sigma_{rr}$ is the sum of the marginal variance of the $i^{\text{th}}$ block of unaveraged $\hat{s}$. Thus, sum of the marginal variance of $\tilde{s}$
which is the sum of the $i^{th}$ diagonal $\mathbf{J}_i^t\mathbf{\Sigma}_{ii}\mathbf{J}_i$ is also smaller or equal to the sum total of marginal variance of $\hat{s}$. This implies that
the marginal variance of the posterior mean decreases as a result of the diagonal of the variance matrix shrinking in magnitude
upon averaging.
**C2   Proof of the reduction in model resolution when aggregation is performed**
Aggregated forward operator $\tilde{\mathbf{H}}$ can be written as:
$\tilde{\mathbf{H}} = \mathbf{HB},$ \hfill (C10)
where $\mathbf{B}$ is the upscaling matrix. Dimension of $\mathbf{B}$ has the dimension of transpose of $\mathbf{J}$. Structural form of $\mathbf{B}$ is similar to the
form of $\mathbf{J}$ explained in C2. Non-zero entries of $\mathbf{B}$ are in the same place as $\mathbf{J}'$ with magnitude replaced by unity. This is evident
from the fact that forward operator is summed instead of being averaged for aggregation. Properties of $\mathbf{B}$ are as follows:
1. $\mathbf{B1} = \mathbf{1}$
2. $\mathbf{JB} = diag(\mathbf{N})^{k\times k}$  where $\mathbf{N}$ is the vector of number of neighboring grid-cells for any particular grid-cell i.e. $\mathbf{N} = (N_1,\ldots,N_k)$
3. $\mathbf{BJ} = \begin{bmatrix} \mathbf{C_1} & \mathbf{0} & \dots & \mathbf{0} \\ \mathbf{0} & \mathbf{C_2} & \dots & \mathbf{0} \\ \vdots & \vdots & \ddots & . \\ \mathbf{0} & . & \dots & \mathbf{C_k} \end{bmatrix}^{m \times m}$    is a block diagonal matrix. Any block $\mathbf{C}_i$ of $\mathbf{JA}$ can be expressed as a varying di-
mension (depending on the number of neighboring grids of any particular grid-cell) matrix of form:
$$\mathbf{C}_i = \begin{bmatrix} \frac{1}{N_i} & \cdots & \frac{1}{N_i} \\ \vdots & \ddots & \vdots \\ \frac{1}{N_i} & \cdots & \frac{1}{N_i} \end{bmatrix}^{N_i \times N_i} = \frac{1}{N_i} \mathbf{1} \mathbf{1}^t \tag{C11}$$

4. $\mathbf{BJ}$ is symmetric and positive semi-definite
First three properties are simple observations from the construction. So, here we provide proof of the fourth property.
*Proof.* By construction, $Det(\mathbf{BJ} - \lambda \mathbf{I}) = Det(\mathbf{C_1} - \lambda \mathbf{I})\dots.Det(\mathbf{C_k} - \lambda \mathbf{I})$. So, eigenvalues of $\mathbf{BJ}$ are the list of eigenvalues
of the block matrices. It can be proved that 1 and 0 are the only two distinct eigenvalues of $\mathbf{C}_i$ for any $i$. Below here is a brief
argument on that:

$\left(\frac{1}{N_i}\mathbf{1}\mathbf{1}^t\right)\mathbf{1} = \frac{1}{N_i}\mathbf{1}N_i = 1 \cdot \mathbf{1}$ implies one eigenvalue of $\mathbf{C_i}$ is 1. Observe that, $rank\left(\frac{1}{N_i}\mathbf{1}\mathbf{1}^t\right) = rank(\mathbf{1}) = 1$. Hence, dimen-
sion of null space $dim\left(\mathcal{N}\left(\frac{1}{N_i}\mathbf{1}\mathbf{1}^t\right)\right) = k - rank\left(\frac{1}{N_i}\mathbf{1}\mathbf{1}^t\right) = k - 1$. This implies that the other eigenvalue of $\mathbf{C_i}$ is 0 with
multiplicity $k - 1$.

So, not only $\mathbf{C}_i$ is symmetric but also the eigenvalues $\mathbf{C}_i$ are always non negative. Consequently, all eigenvalues of $\mathbf{BJ}$ are
of similar form i.e. $\mathbf{BJ}$ is symmetric positive semidefinite.    $\square$
Finally, model resolution matrix for inversion can be written as $\frac{\partial \hat{\mathbf{s}}}{\partial \mathbf{z}}\mathbf{H}$ where $\mathbf{H}$ is the forward operator operator. Post inversion
aggregated model-resolution can be written as:
$$\frac{\partial \tilde{\mathbf{s}}}{\partial \mathbf{z}}\tilde{\mathbf{H}} = \mathbf{A}\frac{\partial \hat{\mathbf{s}}}{\partial \mathbf{z}}\mathbf{HB} \quad \text{By Eq. (C1) and C10} \tag{C12}$$

The question is what happens to the trace of the model-resolution under the aggregated scenario? We provide a proof for the
simple batch Bayesian case in lemma C2. Proof for the geostatistical case is similar and left for the enthusiastic readers.

**Lemma 1.**

$$\mathbf{Mres} = \mathbf{QH}'\psi^{-1}\mathbf{H}$$

$$\mathbf{Mres}_{aggregated} = \mathbf{JQH}'\psi^{-1}\mathbf{HB} \quad then$$

$trace\left(\mathbf{Mres}_{aggregated}\right) \leq trace(\mathbf{Mres})$ \hfill (C13)

*Proof.* Model resolution for the aggregated scenario can be written as:

$$\text{trace}\left(\mathbf{Mres}_{\text{aggregated}}\right) = \text{trace}(\mathbf{JQH'}\psi^{-1}\mathbf{HB}) = \text{trace}(\mathbf{BJQH'}\psi^{-1}\mathbf{H}) = \text{trace}(\mathbf{WS}) \text{ where } \mathbf{W} = \mathbf{BJ}, \mathbf{S} = \mathbf{QH'}\psi^{-1}\mathbf{H},$$
$$\tag{C14}$$

where $\mathbf{S}$ and $\mathbf{W}$ are both of dimension $(m \times m)$. $\mathbf{S}$ is a positive semidefinite matrix since both $\mathbf{Q}$ and $\mathbf{H'}\psi^{-1}\mathbf{H}$ are positive semidefinite. For $\mathbf{W}^{m \times m}$ and $\mathbf{S}^{m \times m}$ positive semidefinite, trace of their product can be bounded by the following quantities (see Kleinman and Athans, 1968 and discussion in Fang et al., 1994):

$$\lambda_{min}(\mathbf{W})trace(\mathbf{S}) \leq trace(\mathbf{WS}) \leq \lambda_{min}(\mathbf{W})trace(\mathbf{S}) \tag{C15}$$

By Property 4 of the weight matrix $\mathbf{B}$, we know that $\lambda_{min}(\mathbf{W}) = 0$ and $\lambda_{max}(\mathbf{W}) = 1$, hence the above reduces to $0 \leq trace(\mathbf{WS}) \leq 1 \cdot trace(\mathbf{S})$. Hence is the proof by C14.

□

*Author contributions.* V.Y., and S.G. contributed equally in preparing the manuscript.

*Competing interests.* The authors declare no competing interest.

*Acknowledgements.* The authors thank Anna Karion, Kimberly Mueller, James Whetstone (National Institute of Standards and technology, NIST), and Daniel Cusworth (University of Arizona, UA) for their review and advice on the manuscript. This work was partially funded by NIST's Greenhouse Gas Measurements Program. Support to University of Notre Dame provided by NIST grant 70NANB19H132. Support for JPL was provided via an interagency agreement between NIST and NASA. A portion of this research was carried out at JPL, California Institute of Technology, under a contract with NASA (80NM0018D0004).

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
