# Peer review of "Metrics for evaluating the "quality" in linear atmospheric inverse problems: a case study of a trace gas inversion"

_Geoscientific Model Development, 2022_

## Author Comment (AC1)

Dear Dr. Rayner:

We thank you for your thorough review of our manuscript. We provide our responses to your comments in italicized text. We partition the text in the general comment section of your review to extract questions raised by you and then answer them in sequential order. We also answer the specific comments in sequential order but we restart the numbering to match with the numbering of the specific comments. The changes made in light of your comments are identified by the label Action and follow the Response label.

**General Comments**

1. **Reviewer Comment:** This paper presents a series of quantities that can be derived from linear inverse theory. Put roughly they are the similarity of footprints (nearly the independence of rows of the Jacobian), the local sensitivity of the result to various inputs and finally a global sensitivity using a first-order Taylor expansion with respect to all inputs.

   Response: *This comment summarizes the study.*

   Action: *No action required.*

2. **Reviewer Comment**: The metrics are potentially useful and some are, to my knowledge, novel.

   Response: *We thank the reviewer in appreciating our effort and the contribution of our work.*

   Action: *No action required.*

3. **Reviewer Comment**: The paper is potentially in scope for AMT though I think it needs more work to make it more relevant to likely readers.

   Response: *We think our manuscript is appropriate for GMD as:*

   - *GMD Journal has previously published papers on atmospheric inverse modeling covering a wide variety of topics, and*

   - *Our manuscript falls within the scope identified by the editorial board of the Journal. Please see our response below:*

   *List of a few papers* *among many that have been published in GMD that deal with atmospheric inverse modeling. Some of these are specifically focused on method development*

   - Cho, T., Chung, J., Miller, S. M., and Saibaba, A. K.: Computationally efficient methods for large-scale atmospheric inverse modeling, Geosci. Model Dev., 15, 5547–5565, https://doi.org/10.5194/gmd-15-5547-2022, 2022.

   - Vojta, M., Plach, A., Thompson, R. L., and Stohl, A.: A comprehensive evaluation of the use of Lagrangian particle dispersion models for inverse modeling of greenhouse gas emissions, EGUsphere [preprint], https://doi.org/10.5194/egusphere-2022-275, 2022.

   - Liu, X., Weinbren, A. L., Chang, H., Tadić, J. M., Mountain, M. E., Trudeau, M. E., Andrews, A. E., Chen, Z., and Miller, S. M.: Data reduction for inverse

modeling: an adaptive approach v1.0, Geosci. Model Dev., 14, 4683–4696, https://doi.org/10.5194/gmd-14-4683-2021, 2021.

- Miller, S. M., Saibaba, A. K., Trudeau, M. E., Mountain, M. E., and Andrews, A. E.: Geostatistical inverse modeling with very large datasets: an example from the Orbiting Carbon Observatory 2 (OCO-2) satellite, Geosci. Model Dev., 13, 1771–1785, https://doi.org/10.5194/gmd-13-1771-2020, 2020.

- Hase, N., Miller, S. M., Maaß, P., Notholt, J., Palm, M., and Warneke, T.: Atmospheric inverse modeling via sparse reconstruction, Geosci. Model Dev., 10, 3695–3713, https://doi.org/10.5194/gmd-10-3695-2017, 2017.

- Miller, S. M., Michalak, A. M., and Levi, P. J.: Atmospheric inverse modeling with known physical bounds: an example from trace gas emissions, Geosci. Model Dev., 7, 303–315, https://doi.org/10.5194/gmd-7-303-2014, 2014.

- Chai, T., Stein, A., and Ngan, F.: Weak-constraint inverse modeling using HYSPLIT-4 Lagrangian dispersion model and Cross-Appalachian Tracer Experiment (CAPTEX) observations – effect of including model uncertainties on source term estimation, Geosci. Model Dev., 11, 5135–5148, https://doi.org/10.5194/gmd-11-5135-2018, 2018.

Is the submitted manuscript within the scope of the journal?

*Our paper covers two focus areas identified by the Journal*

- *Development and technical papers, describing developments such as new parameterizations or **technical aspects of running models** such as the reproducibility of results;*

- *New methods for assessment of models, **including work on developing new metrics for assessing model performance** and novel ways of comparing model results with observational data*

Does it contribute anything new to the field or atmospheric inverse modeling?

*In this study we provide:*

- *Analytical expressions to conduct post hoc (that is after an inversion has been performed) local sensitivity analysis by computing partial derivatives*

- *Demonstrate a scientifically interpretable framework for ranking thousands of spatio-temporally correlated input parameters with same or different units.*

- *A mathematical schema for global sensitivity analysis but it remains considerably harder to perform in the absence of the knowledge about uncertainties associated with all the inputs that go in an inversion.*

- *Develop methods to assess spatio-temporal correlation between forward operators of two or multiple observations. This is tied to overall diagnostics of the estimated fluxes as fluxes remain highly sensitive to the forward operator and improvement in understanding the representation of atmospheric transport through spatio-temporal association in the forward operator can lead to significant improvement in designing the components of a suitable inversion framework.*

*Even though we have a comprehensive awareness of the literature associated with atmospheric inverse modeling and the methods used to assess them but here we would like to refer a paper by Michalak et al., 2017 (see below) that does not discuss sensitivity analysis in the context of linear atmospheric inverse problems*

*Michalak, A. M., Randazzo, N. A., and Chevallier, F.: Diagnostic methods for atmospheric inversions of long-lived greenhouse gases, Atmos. Chem. Phys., 17, 7405–7421, https://doi.org/10.5194/acp-17-7405-2017, 2017.*

Action: *We have modified our manuscript in light of your and second reviewer's comments.*

4. **Reviewer Comment***: My first problem with the paper is its title. The word "assessment" suggests some comment on the quality or robustness of an inversion. The authors don't do that and it's not clear from the paper that the generated metrics can do it.*

Response: *We agree with the reviewer's interpretation about the capability of the metrics that they do not provide any information regarding the robustness of inversion. The choice of the word assessment in the previous title of the manuscript was based on our understanding that providing information about the sensitivity of various inputs that goes into linear atmospheric inverse modeling in governing inverse solution can be considered as an assessment of inversions.*

Action: *After considering reviewers' comments we have now changed the title of the paper as:* **Metrics for evaluating the "quality" in linear atmospheric inverse problems: a case study of a trace gas inversion**

5. **Reviewer Comment***: I'm unclear, for example, what new information is provided by the overlap of footprints. It might well mean that parts of the control space are under-sampled by the observations but the posterior uncertainty already tells us this.*

Response: *We surmise that this comment is with respect to the IOAMI and JSD metrics. Yes, it is true that the both IOAMI and JSD provides information about what parts of the control space is under sampled and we agree that this information can also be obtained from posterior uncertainty, model resolution matrix and the plot of the forward operator or footprints themselves. However, IOAMI and JSD metrics proposed in our paper are more comprehensive as they allow researchers to:*

- *Assess linear or non-linear correlation between footprints in space and time while also accounting for intensity of footprints. This correlation can also be expressed in the units of footprints, which to authors' knowledge is novel in the domain of atmospheric inverse problems.*

- *Build a stable non-stationary covariance model for model-error covariance with diagonal and off-diagonal terms that can be incorporated in the inverse problems. Implementation details about these metrics have been submitted in the code and included in the supplementary material*

- *Note that, although posterior uncertainty can indicate areas of low and high uncertainty, this uncertainty is also conflated with prior uncertainty. Deconvoluting or apportioning this correctly to find out uncertainty contribution*

*from model-data is a challenging problem. In fact, one of the main objectives of the global sensitivity method that we present in our manuscript is to understand this apportionment of posterior uncertainty.*

- o *The technique proposed here does not suffer from this problem and provides an easy way to incorporate this into model-data error covariance. So, to reiterate, here our goal is to not exactly find low/high uncertainty areas but to provide meaningful correlation structure in model-data error covariance.*

Action: *No action required.*

6. **Reviewer Comment**: For the linear Gaussian case the posterior uncertainty can be calculated without measurements.

Response: *Yes, in the linear Gaussian case, we can compute this even if the model-data error covariance is 0. We envision that the reduction term would be very close to prior error covariance with posterior uncertainty being close to 0.*

Action: *No action required.*

7. **Reviewer Comment**: Likewise, the sensitivity of the posterior estimate to the value of a given measurement is potentially useful as a warning flag for measurements that might have undue control on the outcome but it's not really developed.

Response: *The question of the undue importance of observation in governing the estimate of emissions or state vector depends on the goal of the study. In the case study presented in this work, we are interested in knowing the observations that influenced the estimate of emissions at the site of the Aliso Canyon gas leak. In other applications, the goal can be entirely different so providing a method for flagging observation is not desirable. However, the proposed method can rank the importance of observations in governing the estimate of emissions or state vector and section 4.2.1 shows how to evaluate these rankings.*

Action: *Measurement influence via local sensitivity analysis and ranking on posterior estimate is discussed in detail in section 3.4 and 4.2.1*

8. **Reviewer Comment**: The global sensitivity analysis, which allows consideration of all inputs to the linear inverse problem, is potentially more interesting but again is not developed beyond generation of the first-order expansion. The example presents a good opportunity to demonstrate application of these methods but this is not taken far beyond calculation of the diagnostics.

Response: *We thank the reviewer for the comment. The global sensitivity analysis (GSA) presented here uses local sensitivities but actually belongs to the class of variance-based methods. However, regardless of the methods we choose, full GSA is very complicated since it requires knowledge of the first-order joint dependence (aka their covariance) of the parameters. To exemplify, in an atmospheric inversion this would mean knowing the joint dependences of all the parameters including Q, R, and the transport model input parameters. Developing this method beyond first order requires second-order joint dependence (aka third-order joint moments). This is essentially knowing the entire joint distribution of the parameters. Such knowledge is very difficult to obtain in real applications and thus we do not develop it further.*

*As mentioned in the answer of comment 5 of the reviewer, one of the other objectives of the GSA method adopted here is to be able to apportion the posterior uncertainty into Q and R component contributions. We can achieve that via first-order expansion.*

Action: *This section (section 3.3) has been modified to reflect the caveats associated of the proposed method. It has been substantially rewritten in light of your and second reviewer's comments.*

**Reviewer Comment:** I see two possibilities for the paper (1) Repackage it as a technical note focusing on the calculation of the diagnostics or (2) Extend the work to generate diagnostics of overall inversion performance, probably focusing on robustness.

Response: *Our study is a technical note and focuses on the calculation of the proposed diagnostics referred by the reviewer as the first possibility.*

Action: *We have clarified the contribution of our paper in terms of a technical note in the last paragraph of the introductory section of the manuscript. Note as part of the technical note we have provided two detailed MATLAB Livescripts that implements all the diagnostics we provide in the manuscript and this has been mentioned in the last paragraph of the beginning of the section 2 (line 98 to 102).*

**Specific Comments:**

1. **Reviewer Comment:** What is a footprint-induced probability distribution?

   Response: *When we have many realizations of a random variable (scalar or vector-valued), we can compute a probability distribution based on the values of the random variable. This is often called probability distribution induced by the random variable. Likewise, probabilistically if we consider a set of all footprints from a transport and dispersion model as realizations of any underlying random vector, we can also come up with a probability distribution obtained by the footprints also known as a footprint-induced probability distribution.*

   Action: *We acknowledge that this is too pedantic and therefore we have modified this sentence in the revised manuscript (see lines 170-171).*

   **Reviewer Comment:** L190: The definition of the averaging kernel is true but this is an odd motivation for it, much better below when contemplating sensitivity of result to prior

   Response: *We agree that interpreting averaging kernel via local sensitivity of the estimated fluxes with respect to observations is not traditionally done. However, here our goal is simply to establish the link between local sensitivity with respect to observations and ubiquitous averaging kernel and show that averaging kernel and DOFS are just two subcases from the whole spectrum.*

   Action: *No action required.*

   **Reviewer Comment:** Eq. 19: it's worth noting that this sensitivity is very close to the proportional uncertainty reduction A P^-1 and hence the averaging kernel. By the way I thank the authors for making me think hard enough about the relationship between AK, DOFS and uncertainty reduction to finally get an intuitive sense of it

Response: *We convey sincere thanks to the reviewer for this comment. Yes, this sensitivity can also be thought as the proportion of posterior uncertainty to that of the prior uncertainty (i.e. VQ^-1) which intuitively makes sense. Whereas proportional uncertainty reduction is nothing but the averaging kernel. Thus sensitivity w.r.t. prior is negatively correlated with unknown fluxes or averaging kernel.*

Action: *In light of your comment we have added text the manuscript on this subject. Please see lines 254-256 in the revised manuscript.*

2. **Reviewer Comment:** L275: When commenting on covariances between H, Q etc we should also note that constraints like conservation of mass introduce strong covariances within the parameters of H. Covariances can only occur on physically plausible manifolds. This is a profoundly under-studied problem in transport modeling and there is probably great insight to be borrowed from Numerical Weather Prediction.

Response: *We thank the reviewer for this insight. Yes, covariance within H parameters can be significantly high when conservation of mass is a constraint and in certain conditions (i.e. physically plausible manifolds) Q and H parameters can exhibit high correlation. Unless we have good knowledge of these cases, it is not possible to compute these dependencies.*

Action: *No action required.*

**Reviewer Comment:** L431: not sure what the authors mean by aggregation error here. If they're truly commenting on temporal aggregation error they should cite DOI:10.5194/acp-11-3443-2011.

Response: *Yes, we are commenting on the aggregation error.*

Action: *We have now included the reference mentioned by the reviewer.*

---

## Author Comment (AC2)

Dear Reviewer:

We thank the reviewer for his thorough review of our manuscript. We provide our responses to reviewers' comments in italicized text. We answer all the questions in sequential order. However, for clarity we follow separate numbering sequence for Major comments and Specific comments section. The reviewer finally reiterates his questions raised earlier in section Major comments, the response to these comments are provided in the section titled as Final comments and follow separate numbering sequence. The changes made in light of reviewer's comments are identified by the label Action and follows the Response label.

**Major Comments**

1. **Reviewer Comment:** I couldn't fully understand the contributions of this paper. Is it claiming to be developing new methods for inverse problems, or applying existing techniques to the case study? The methods for local sensitivity analysis involving derivatives appear to be a special case of the Hyperdifferential sensitivity analysis for linear inverse problems and the GSA method is essentially DGSM (see end of review).

   Response: *Contribution of the Paper*

   *In this study, we focus on linear atmospheric inverse problems and provide:*

   - *Analytical expressions to conduct post hoc (that is after an inversion has been performed) local sensitivity analysis (LSA) by computing partial derivatives*
   - *A scientifically interpretable framework for ranking thousands of spatio-temporally correlated input parameters with the same or different units.*
   - *A mathematical schema for global sensitivity analysis (GSA). However, it remains harder to perform GSA in the absence of knowledge about uncertainties associated with all the inputs that go in an inversion.*
   - *Methods to assess spatio-temporal correlation between forward operators of two or multiple observations. This is tied to overall diagnostics of the estimated fluxes as fluxes remain extremely sensitive to the forward operator and improvement in understanding the representation of atmospheric transport through spatio-temporal association in the forward operator can lead to significant improvement in designing the components of a suitable inversion framework.*

   *Even though we have a comprehensive awareness of the literature associated with atmospheric inverse modeling and the methods used to assess them but here we would like to refer to a review paper by Michalak et al., 2017 (see below) that does not discuss sensitivity analysis, or the other two contributions mentioned above.*

   Reference:

   - Michalak, A. M., Randazzo, N. A., and Chevallier, F.: Diagnostic methods for atmospheric inversions of long-lived greenhouse gases, Atmos. Chem. Phys., 17, 7405–7421, 2017.

   Response: *Is it claiming to be developing new methods for inverse problems?*

*We do not claim that we are developing new methods for understanding the sensitivity of inputs for all classes of inverse problems and nowhere is it mentioned in the previous or revised versions of the manuscript. Having said that in this work we do not specifically borrow or apply any existing method designed for dealing with parameter sensitivity. We agree that our method falls within the same category as Bayesian Hyperdifferential sensitivity analysis (BHDSA). However, the way we conduct sensitivity analysis and rank the importance of inputs is different from what is proposed in the BHDSA manuscript.*

*To support our argument, we highlight the following properties of linear atmospheric inverse problems, which necessitates a tailored approach for conducting sensitivity analysis.*

*Properties of Atmospheric Inverse Problems*:

- *Most of the input parameters (this includes measurements) are correlated in both space and time with correlation dependent on the weather conditions like the direction of the wind, rainfall, temperature, boundary layer height, clouds, and solar radiation among many others.*

- *Inputs to an atmospheric inverse problem result in a dense vector or matrix (depending on how the output is arranged) containing millions of entries depending on the spatio-temporal resolution of the problem. Furthermore, spatio-temporal correlation structure of inputs is different across observational platforms like satellites, aircrafts, and in-situ sites, among others. This means that the spatio-temporal correlation between observations collected from one observational platform is different from those obtained from another platform and there also exists cross-correlation across these observations (see Figure 1).*

[Figure]

× OCO-2 Target Mode Observations
• CLARS-FTS Observations
▲ IN-SITU Observations
▇ Los Angeles Basin Boundary

**Figure 1**. *Observation locations of carbon dioxide and methane concentrations obtained from different platforms in Los Angeles on 8/5/2015.*

- *Output in some cases can only have non-negative or zero values in space and time.*

- *The ranking of inputs should be preferably bounded, scientifically interpretable, and consistent.*

*Due to these properties of atmospheric inverse problems some of the previously proposed approaches (we do not list all) cannot be directly used in their current form.*

- *One at a time or Moris method for ranking.*

- o *Space-time correlation is not accounted for in the method.*

- o *Even if we assume independence of inputs the approach is computationally infeasible for thousands of inputs. If we assume that the output results in a matrix or vector of 10 thousand entries and only 10 samples (which is a large underestimate) are used to cover the sampling space of each parameter and it takes 1 minute to do an inversion on a supercomputer (which is also a large underestimate) then it would take months to obtain the relative ranking of inputs.*

- ▪ *Monte Carlo-based methods: Sobol Class*

  - o *See the same computational rationale as above. Sampling from the complex spatio-temporal structure to compute high-dimensional integral directly or via Monte Carlo integration with thousands or millions of parameters would mean that we would be sampling for a long time.*

- ▪ *FAST method*

  - o *Has a similar problem of sampling from complex spatio-temporal structure as mentioned above. Furthermore, in atmospheric inverse problems, we need something that is localized in both space and time whereas FAST is only localized in frequency space. Wavelets have the property of being localized in space and time and may be utilized to compute FAST-based sensitivities. However, we do not investigate how different kinds of wavelets for different inputs, units, constraints, and space-time correlation structures will work for sensitivity analysis. Furthermore, there is no previous research to support that it has been done and whether it is computationally feasible to do this for large number of inputs.*

- ▪ *Polynomial Chaos Expansion (PCE)*
  - o *Does not work. Joint pdf of inputs is not known. See below given reference for the methodological constraints associated with PCE.*

    *Reference:*

    *Sudret, Bruno. "Global sensitivity analysis using polynomial chaos expansions." Reliability engineering & system safety 93.7 (2008): 964-979.*

- ▪ *Derivative-based Global Sensitivity Measures (DGSM)*

    *We agree that DGSMs can also be applied under these scenarios. However, neither the method proposed and demonstrated in this work nor DGSM can be deployed for conducting GSA in atmospheric inverse problems except for the cases where number of parameters of interest is small and their covariance structure is known. We show a realistic example of this in the manuscript with $Q$ and $R$ parameters which is contextually highly relevant and their covariance can also be estimated under certain assumptions. However, a generic application of this method is not possible since covariance between spatio-temporally correlated parameters (note we refer to observations as parameters) is unknown in these applications. The method*

*we proposed for conducting global sensitivity analysis (GSA) in this study is not exactly DGSM. It is a Taylor's series based approximate method for conducting GSA. However, this is not really an assumption free method. It assumes that covariance (or at least their variances) between the parameters can be obtained. It does belong to the variance-based decomposition class of GSA and requires partial derivatives.*

- *Bayesian Hyperdifferential Sensitivity Analysis (BHDSA)*

  o *We thank the reviewer for directing our attention to the BHDSA article. We acknowledge that the local sensitivity analysis (LSA) approach described in our manuscript falls in the same category as BHDSA. However, we utilize analytical derivatives of the closed form MAP solution with respect to the input parameters in atmospheric linear inverse problems to conduct LSA. Therefore, we do not need to evaluate discretized sensitivity operator (described in the BHDSA manuscript) since we know the exact functional form of the MAP solutions under the setup assumed in this manuscript. Thus, we avoid generating data samples for conducting multiple inversions.*

Action: *We have now included references and added a couple of paragraphs in the introductory section of the manuscript explaining why existing methods would not work for in the context of atmospheric inverse problem. Please also see our responses in the specific comments section.*

2. **Reviewer Comment: Mathematical exposition**: I understand that this is not an applied math journal, but the standard for exposition was well below this journal.

Response: *This is a subjective statement and from our side, we can only make sure that every symbol used in equations is properly defined and equations are clear and demonstrated by an implementation, which we have also done through submitted MATLAB Livescripts.*

*It would be quite useful for us to know from the reviewer the standard for mathematical exposition for this journal so we can follow it.*

*In our work, some of the equations are long where many matrix multiplication and inversion terms are repeated and therefore, we have replaced them by a single symbol that shortens the equations, makes them legible, and amenable for easy implementation in any programming language. This also reduces the computational cost as repeated operations are avoided.*

*We cannot provide context of every equation and how they should be used. We have followed mathematical formulation for the linear atmospheric inverse problems as used in the field for the past three decades in several papers some of which are published in this Journal as mentioned below.*

References:

- *Miller, S. M., Saibaba, A. K., Trudeau, M. E., Mountain, M. E., and Andrews, A. E.: Geostatistical inverse modeling with very large datasets: an example from the Orbiting Carbon Observatory 2 (OCO-2) satellite, Geosci. Model Dev., 13, 1771–1785, https://doi.org/10.5194/gmd-13-1771-2020, 2020.*

- *Hase, N., Miller, S. M., Maaß, P., Notholt, J., Palm, M., and Warneke, T.: Atmospheric inverse modeling via sparse reconstruction, Geosci. Model Dev., 10, 3695–3713, https://doi.org/10.5194/gmd-10-3695-2017, 2017.*

- *Miller, S. M., Michalak, A. M., and Levi, P. J.: Atmospheric inverse modeling with known physical bounds: an example from trace gas emissions, Geosci. Model Dev., 7, 303–315, https://doi.org/10.5194/gmd-7-303-2014, 2014.*

- *Yadav, V. and Michalak, A. M.: Improving computational efficiency in large linear inverse problems: an example from carbon dioxide flux estimation, Geosci. Model Dev., 6, 583–590, https://doi.org/10.5194/gmd-6-583-2013, 2013.*

Action: *We have checked for all the instances of the issues mentioned in section 3's comment 3 and made the necessary changes. These are listed under section 3's comment 3.*

3. **Reviewer Comment:** The notation is not setup properly and inconsistently used.

    Response: *We request the reviewer to provide examples so that we can address his concern.*

    Action: *We have checked our manuscript for consistency of symbols and corrected them for errors and wherever required provided more context and explanation of the symbols.*

4. **Reviewer Comment:** There were a lot of vague statements (I did not fully tabulate this list).
    Response: *We request the reviewer to provide specific examples so that we can address his concern. We have responded to all the reviewer's concerns with respect to the issues raised under specific comments section. Please also see the above-mentioned actions (see comment 3 under major comments) we have taken considering the reviewer's comments.*

    Action: *No action required.*

**Specific Comments**

1. **Reviewer Comment**: Title: I think working in the word sensitivity is better here than "assessing".

    Response: *We deliberately did not use the word sensitivity as sometimes the forward operator/ forward model in linear atmospheric inverse problems is referred to as Jacobian which contains sensitivity of observations to emissions. Furthermore, this paper also deals with deriving correlation functions and therefore just using the word sensitivity reduces the scope of the paper. Thus, we think these are metrics for evaluating the quality of the solution of the linear atmospheric inverse problems.*

    Action: *Regardless, we have changed the title of the manuscript to: **Metrics for evaluating the "quality" in linear atmospheric inverse problems** as this covers overall theme of our study.*

2. **Reviewer Comment**: The abstract does not clarify, of what quantity (i.e. MAP estimate) the sensitivity is being computed. GHG is not expanded and Jacobian should be capitalized throughout.

Response: *Agreed*

Action: *We have revised the abstract and added "estimated fluxes" as the quantity (see line 5 and 12 of the abstract) for which the sensitivity is being computed. We have removed the word GHG from the abstract.*

3. **Reviewer Comment**: The word Jacobian (to me) is misleading because this is a linear problem – there are other Jacobians used in the paper.

   Response: *We are not exactly clear what the reviewer meant by "there are other Jacobians used in the paper". There is only one definition of the Jacobian in our previous manuscript as described in section 3.2. However, for the sake of clarity, we have now removed the word "Jacobian" from the revised manuscript.*

   Action: *In the revised manuscript we have defined Jacobian (used only at one place in the manuscript) in the context of linear atmospheric inverse problems in section 3.1, i.e., before we start describing our metrics for assessing the Jacobian. Note we do not use the word Jacobian in the revised manuscript anymore to avoid confusion as pointed out by the reviewer. We have replaced it with the more general term i.e., forward operator. The forward model/operator in the context of this study (linear atmospheric inverse problems) has a specific meaning which is not applicable in the case of all inverse problems. In the previous version of the manuscript by Jacobian, we meant the forward operator which encodes the sensitivity of observations to the sources of emissions obtained from an atmospheric transport model. For description see:*

   Reference:

   - *Michalak, Anna M., Lori Bruhwiler, and Pieter P. Tans. "A geostatistical approach to surface flux estimation of atmospheric trace gases." Journal of Geophysical Research: Atmospheres 109.D14 (2004).*

4. **Reviewer Comment**: The introduction does not clearly list the contributions (as mentioned earlier) and is missing some references.

   *Response*: *Agreed*

   Action: *We have now included two paragraphs in the introduction listing previous works on sensitivity analysis and their applicability and lacuna for conducting sensitivity analysis in the linear atmospheric inverse problems.*

**Section 3.**

1. **Reviewer Comment**: The inversion is not setup properly before 3.1.1-3.1.3 are explained. I essentially did not understand anything in these subsections. I am not sure what sets are being considered, how this relates to the inversion, etc. There is a small discussion at the end of 3.1.3 but it is referring to things that haven't yet been defined.

   Response: *A linear trace gas atmospheric inverse problem requires many inputs before an inversion can be performed. Among these inputs is an output from a transport model (in*

*generic terms a forward operator or model) that consists of sensitivities of observations to emissions (called as **H** in the manuscript). An example of these sensitivities is shown in Figure 2 for some of the sites included in this case study.*

*Here, the term "sensitivity" is not to be confused with sensitivity analysis.*

[Figure]

**Figure 2**. *Example footprints or sensitivities of observations to emissions (**H**) for a selected location in space, time and altitude above ground level.*

*These sensitivities are essentially 5d pulse functions indexed by latitude, longitude, altitude, and time with sensitivity serving as the $5^{th}$ dimension (see Figure 2). An example 2d plot of this pulse function (for fixed altitude, and time) for the ONT site is shown in Figure 3. When we assess the correlation between two of these pulse functions in terms of shared sensitivity of observations to emissions (forward operator/map or Jacobian **H**) then it can be expressed or defined on a set of grid points as done in the previous version of the manuscript in section 3.1.1*

[Figure]

***Figure 3***. *Sensitivity of observation taken at time 2015-10-23-19-00 to methane emissions for ONT site with coordinates 34.064N and -117.583W located at altitude 41 m above ground level is shown below. Note altitude is fixed in Figure 3.*

*We do not need an inversion to obtain a forward operator for a particular spatio-temporal location and altitude (above sea level) on Earth's surface (see Shiga et al. (2013) and Lopez-Cotez (2017) for creating **H** for hypothetical locations before performing inversion). Section 3.1.1 and 3.1.2. focus on quantifying the correlation between sensitivities of observations to emissions in the forward operator **H**. These correlation functions are based on distance metrics that consider the closeness of sensitivities of observations to emissions in space and time. To the authors' knowledge, such kind of scientifically interpretable correlation functions have not been defined in the context of linear atmospheric inverse problems which can also be expressed in the units of **H** like in Figure 2&3 (Note units can be different in different inverse problems).*

*We can use these correlation functions to define a correlation matrix shown in Figure 4. This correlation matrix is based on the correlation computed through IOAMI and JSD for locations shown in Figure 2. This can then be used as one of the components of model-data error covariance in **R** (also see MATLAB Livescript in the supplementary material) in an inversion. The choice of such components of **R** can have a significant impact on the **performance of inversions** (see Locatelli et al. 2013). This is what is mentioned at the end in section 3.1.3. Note that such analysis of **H** through correlation functions can be done independently even without performing an inversion.*

[Figure]

***Figure 4***. *Example of the correlation matrix based on IOAMI. Note it consists of autocorrelation for observations collected at different time at same locations and cross correlation between observations collected at different sites.*

References:

- *Lopez-Coto, Israel, et al. "Tower-based greenhouse gas measurement network design—the national institute of standards and technology north east corridor testbed." Advances in Atmospheric Sciences 34.9 (2017): 1095-1105.*

- *Shiga, Yoichi P., et al. "In-situ CO2 monitoring network evaluation and design: A criterion based on atmospheric CO2 variability." Journal of Geophysical Research: Atmospheres 118.4 (2013): 2007-2018.*

- *Locatelli, R., et. al., Impact of transport model errors on the global and regional methane emissions estimated by inverse modelling, Atmos. Chem. Phys., 13, 9917–9937, https://doi.org/10.5194/acp-13-9917-2013, 2013.*

Action: *Considering the reviewer's concerns we have now included a flowchart and defined all the components of atmospheric inverse models that we have used in the manuscript (see Figure 1 in the revised manuscript). We have simplified the definition of the set used in section 3.1.1. In the revised manuscript they are described in section 3.1.1 (line number 128-130).*

2. **Reviewer Comment**: Section 3.2: Someone not familiar with this material will struggle since it has not been discussed properly. I suggest reorganizing this section bringing some of this material earlier.

   Response: *In the previous version of this manuscript, we have referred to other previous studies on linear atmospheric inverse problems, which provided enough background of the setup we use (see a reference given below). Therefore, we do not describe the background to inverse problems in detail in section 3.2.*

   Action: *We have now included a brief background paragraph in the organization section (Section 2) and provided a diagrammatic description of atmospheric inversions (see Figure 1 in the revised manuscript). This is mostly a modified version of figure 1 in Gourdji et al. 2008.*

   Reference:

   - *Gourdji, Sharon M., et al. "Global monthly averaged CO2 fluxes recovered using a geostatistical inverse modeling approach: 2. Results including auxiliary environmental data." Journal of Geophysical Research: Atmospheres 113.D21 (2008).*

3. **Reviewer Comment**: Notation/Writing: This is not consistent throughout the paper. Sometimes subscripts refer to sizes, other times they mean elements. Sometimes boldfaced, sometimes not. Sometimes lower/upper case. Sentences should not start with variables. When units are being described, the entries of vectors have units, not the vectors themselves.

   Response*: We thank the reviewer for this comment.*

   Action: *We have corrected the inconsistencies that the reviewer mentioned here along with the ones we found.*

4. **Reviewer Comment**: Line 157: it's -> its. This occurred in other places also

   *Response*: We thank the reviewer for pointing this out.*

   Action: *We have fixed this error in the revised manuscript.*

5. **Reviewer Comment**: Line 169: it can be simplified using the notation below in 172.

   Response*: We thank the reviewer for pointing this out.*

   Action*: We have simplified the expression in line 214 i.e., Eq. (13) (line 169 in the previous manuscript) in the revised manuscript.*

6. **Reviewer Comment**: Line 173: One differentiates a function rather than an equation.

   Response*: Right. However, often a function is represented as an equation. Therefore, we can differentiate an equation. Of course, we understand that the converse may not be true. An equation may simply be tautologies. However, in the context of atmospheric inversions, often these functions are referred to as equations and we tried to stay close to their nomenclatures. We refer the reviewer to the definition of a function mentioned on a university webpage. See below.*

   *https://tutorial.math.lamar.edu/classes/calci/Functions.aspx*

   Action: *No action required.*

7. **Reviewer Comment**: When referring to equations, one typically puts a parenthesis, e.g. (10).

   Response*: We thank the reviewer for making this point.*

   Action: *We have fixed this in the revised manuscript.*

8. **Reviewer Comment**: Line 178: The inputs to the inverse problems are data and hyperparameters. The transport model is typically fixed, as is the drift matrix X. What does it mean to differentiate wrt them? Why is that useful in applications? Are these matrices functions of some hyperparameters?

   Response: *In atmosphere inverse problems (which may be true for other inverse problems) we want to holistically know the impact of all inputs that went into governing the solution of the inverse problem. This would also include $\mathbf{X}$ and the forward operator or transport model.*

   ***What does it mean to differentiate wrt $\mathbf{X}$ and $\mathbf{H}$?***

   *Differentiating with respect to X and the transport model implies evaluating the impact of entries in these matrices on the inverse estimates of emissions. Below here we describe their meaning in the context of linear atmospheric problems:*

   ***Why we need to differentiate with respect to entries in a fixed transport model?***

   *Atmospheric transport: Established models do a relatively good job in modeling atmospheric transport on flat terrain under stable weather conditions but do an extremely poor job in complex terrain and mountains. Furthermore, if the winds are highly variable then this poses another challenge (e.g. predicting wind gusts vs average wind speed). Technically, all these errors should be accounted for in the model-data error covariance but these are mostly mis-specified (see a selected reference below for the impact of misspecification; Note, this literature is long and there are many articles only on this*

*particular problem) and their consequences can be extremely serious. In these situations, we want to understand the impact of atmospheric transport on the inverse solution vis-à-vis other inputs.*

*In the case study presented in this work, Figure 4 subplot B shows atmospheric transport associated with the most important observation. A slight shift in the location of highest intensity in subplot B can be quite serious as we can have a situation where we have an observation that indicates a high concentration of greenhouse gas but the transport predicts the spatial location to be miles (just an example) away. In these circumstances, we would end up assigning emissions to a location, which probably did not even have any sources to begin with. Thus, knowledge of the sensitivity of fluxes to transport can be very useful here. This can also help in better construction of model-error covariance.*

*Reference: Locatelli, R., et. al., Impact of transport model errors on the global and regional methane emissions estimated by inverse modeling, Atmos. Chem. Phys., 13, 9917–9937, https://doi.org/10.5194/acp-13-9917-2013, 2013.*

**Why we need to differentiate with respect to entries in a prior or $X$?**

*A misspecification of prior or $\mathbf{X}$ can also lead to disastrous consequence on the estimate. For mathematical justification see:*

Reference:

- *Nguyen, H.; Cressie, N.; Hobbs, J. Sensitivity of Optimal Estimation Satellite Retrievals to Misspecification of the Prior Mean and Covariance, with Application to OCO-2 Retrievals.* Remote Sens. ***2019***, 11, 2770. [https://doi.org/10.3390/rs11232770](https://doi.org/10.3390/rs11232770)

*Note atmospheric inverse problems sometimes are highly underdetermined where observation to state vector ratio can be even lower than 1:100. In such a scenario if we are wrong with the prior then posterior updates can lead to completely incorrect conclusions. We would not go into further details as the reference given above covers it in detail.*

*Finally, sometimes in atmospheric inverse problems, we also want to be extremely cautious and like to analyze the impact of every input to an inverse problem as the answer to the question can have significant health and economic implications.*

*Reference:https://ww2.arb.ca.gov/sites/default/files/2020-07/aliso_canyon_methane_emissions-arb_final.pdf*

**Are these matrices functions of some hyperparameters?**

*In our case no. However, almost all of the inputs can have hyperparameters. This is especially the case in the Hierarchical Bayesian framework. We can also consider a prior at a particular location in space and time to have hyperparameters in some cases.*

9. **Reviewer Comment**: Line 185: The equation is not referred to as Lambda but one of the terms there is Lambda.

Response: *We refer to this as Lambda in the GMDD version of this manuscript as in previous works (see below-given references) it has been referred to as Lambda.*

*We did not find any place in the GMDD version of the manuscript where we used Lambda as one of the terms.*

References:

- *Michalak, Anna M., Lori Bruhwiler, and Pieter P. Tans. "A geostatistical approach to surface flux estimation of atmospheric trace gases." Journal of Geophysical Research: Atmospheres 109.D14 (2004).*

- *Gourdji, S. M., et al. "Regional-scale geostatistical inverse modeling of North American CO 2 fluxes: a synthetic data study." Atmospheric Chemistry and Physics 10.13 (2010): 6151-6167.*

- *Gourdji, Sharon M., et al. "Global monthly averaged CO2 fluxes recovered using a geostatistical inverse modeling approach: 2. Results including auxiliary environmental data." Journal of Geophysical Research: Atmospheres 113.D21 (2008).*

Action: *To avoid confusion we have removed Lambda term in the equation*

7. **Reviewer Comment**: Line 203: When differentiating a vector wrt a matrix, one should get a tensor. I think what is happening is that you are vectorizing X and then differentiating. But the notation is not clear. Same with line 227.

Response: *We do get a tensor. We chose to write the equation in a Kronecker form (see page 2 last paragraph in Sören & Giesen 2018 that says that tensor and Kronecker form are equivalent). This is demonstrated in the code that is attached as a supplementary material with the manuscript. This mathematical formulation is also described in Lutkepohl 1997 and Magnus and Neudecker 2019.*

References:

- *Laue, Sören, Matthias Mitterreiter, and Joachim Giesen. "Computing higher order derivatives of matrix and tensor expressions." Advances in neural information processing systems 31 (2018).*

- *Lutkepohl, Helmut. "Handbook of matrices." Computational statistics and Data analysis 2.25 (1997): 243.*

- *Magnus, Jan R., and Heinz Neudecker. Matrix differential calculus with applications in statistics and econometrics. John Wiley & Sons, 2019.*

8. **Reviewer Comment**: I agree that full GSA is very complicated. It's not clear how this is GSA. Is this essentially derivative based global sensitivity? Once again, I did not really understand what was being discussed.

Response: *We thank the reviewer for the comment. The global sensitivity analysis (GSA) presented here leverages local sensitivities but actually belongs to the class of variance-based methods. It is derivative based but it doesn't require partial derivatives over the entire parameter space (Derivative based Global Sensitivity measures, Sobol et al., 2010). This is an approach that addresses the contribution to the total variance of the estimated fluxes. We can say that this is an approximate method unlike the exact decomposition method of Sobol using conditional variances. It applies a simple first-order Taylor's*

*approximation around parameter estimates to obtain an approximate representation. This approach has been used previously by many researchers (see Hamby et al., 1994, Groen et al., 2017 and Heijungs, R 1996).*

*While we acknowledge that GSA can be done in several ways, e.g. by Sobol's variance decomposition method, and via Derivative based global sensitivity measures (DGSM; Constantine et al.), it is prohibitively expensive to directly apply Sobol's method or DGSM (owing to the difficulty of computing high-dimensional integrals). However, we acknowledge that both DGSM and the method we adopt here can be used when the number of parameters of interest is small and their covariance structure is known. We show a realistic example of this in the manuscript with Q and R parameters which is contextually highly relevant and their covariance can also be estimated under certain assumptions. However, a generic application of this method is not possible since covariance between spatio-temporally correlated parameters (note we refer to observations as parameters) is unknown in these applications. This is also mentioned above in response to reviewer's first comment.*

*We thank the reviewer for mentioning the active-subspace method. In atmospheric inverse problems, it is not possible to directly compute active subspaces owing to the fact that it is expensive to compute empirical covariance of the gradient vector. Empirical covariance requires samples of the gradient vectors at a chosen point. Choosing these points in high-dimensional space requires knowledge of the distribution of high-dimensional parameter vectors and requires running the inversion multiple times. Furthermore, it is difficult to know the joint probability distribution of the parameter space (i.e., knowing the joint dependence of the prior-error variance, model-data error variance, transport model parameters, etc.). In fact, this is one of the reasons, we couldn't fully apply Taylor's series-based method.*

*We realize that it might be possible to leverage the closed-form analytical expressions of the gradients, prior domain knowledge of the parameters, or by other computational means to compute empirical covariance of the gradient vectors along with partial SVD to obtain the first few eigenvectors. In this paper, in the GSA section, our goal is to show how an approximate decomposition/apportionment of the uncertainties can be obtained with respect to the prior error and the model data error variance parameters when we don't know their joint dependence. If we can compute their joint dependences, then we will be able to fully apply this and also the active subspace method. Note that, regardless of the methods, it is extremely challenging to estimate the joint variation of the transport model components, prior-error variance, and model-data error variance components.*

Action: *We have now included a paragraph in section 3.3 and edited introduction and discussion to describe this.*

References:

- *Kucherenko, S. (2010). A new derivative-based importance criterion for groups of variables and its link with the global sensitivity indices. Computer Physics Communications, 181(7), 1212-1217.*

- *Hamby, D. M. (1994). A review of techniques for parameter sensitivity analysis of environmental models. Environmental monitoring and assessment, 32(2), 135-154.*

- *Groen, E. A., Bokkers, E. A., Heijungs, R., & de Boer, I. J. (2017). Methods for global sensitivity analysis in life cycle assessment. The International Journal of Life Cycle Assessment, 22(7), 1125-1137.*

9. **Reviewer Comment**: The authors raise a good point here cautioning against sensitivity metrics since they are of different units but little has been done to address them. There are techniques in sensitivity analysis that the authors should consult (DGSMs and activity scores are some techniques, see Constantine and Diaz)

    Response: *Please see our explanation in paragraph 4 of comment 8 in section 3.*

    *In this manuscript, we use a variation of multiple linear regression to rank sensitivities. The section 3.4 of the manuscript provides a clear description of the methodology that we adopted to rank these sensitivities. Our methodology can compute the rank of parameters with same and different units. For more details see answer to comment 5 in Final comment section.*

    Action: *To address this question we have included explanation in the introduction and in section 3.3.*

10. **Reviewer Comment**: I didn't understand the figures since sometimes the sizes of the quantities are not clear. Are entrywise sensitivities being plotted?

    Response: *Yes, entry-wise quantities are being plotted in the figures.*

    Action: *We have now mentioned the dimensions of the matrices that are being plotted in the main text (e.g. see line 398-399 in the revised manuscript).*

11. **Reviewer Comment**: Appendix: The notation was especially problematic here.

    Response: *It is not clear from the reviewer's comment about the places in the previous version of the manuscript where the reviewer encountered notational inconsistencies.*

    Action: *In the revised version, we have corrected all the inconsistencies that we found in the previous version of the manuscript.*

**Final Comments**

1. **Reviewer Comment**: Title I don't think the paper is novel in the methodology, but as an application to atmospheric inverse problems I think it has potential.

    Response: *From the outset, the paper is written from the perspective of linear atmospheric inverse problems and we have actually specified that in the first paragraph of the manuscript. We do not claim that our method is applicable to all different kinds of inverse problems.*

    Action: *In the revised manuscript, we have edited the abstract to reflect the scope of the manuscript.*

2. **Reviewer Comment**: In its current version, I don't think it is worthy of publishing. In addition to the comments I raised, I would encourage the authors to think of the

following questions to improve the utility. What is the computational cost of these approaches?

> Response: *Please see all the answers to the previous questions that address the reviewer's previous concerns. We are aware of the computational cost of the previously suggested different methods to assess the sensitivity of inputs.*
>
> *Atmospheric inverse problems have unique properties which make existing methods untenable. These properties are described at the beginning of our responses. Doing GSA (except in the case of covariance parameters) is computationally extremely expensive and not possible that is why we decided to use LSA to understand the input factors which influence the inverse estimate of the emissions. The computational cost to obtain partial derivatives is minimal in comparison to other methods and is bounded by the computational cost to perform matrix multiplications which at max is $O(n^3)$. Storage issues can arise in which case ij form of equations can be used to compute partial derivatives. Note computation of partial derivatives is just a calculation of analytical expressions mentioned in the manuscript. The main goal is to understand what is influencing (ranking of importance of factors) the grid-scale inverse estimates of the fluxes in space and time.*
>
> Action: *The discussion with respect to the computational cost associated with the approach for sensitivity analysis proposed in this work is given in the Discussion-paragraph beginning from line 517 in the revised manuscript.*

3. **Reviewer Comment**: Can they be computed when the forward model is not available entrywise?

> Response: *No. As mentioned in the first paragraph of the manuscript the analytical expressions are simply not applicable.*
>
> Action: *No action required.*

4. **Reviewer Comment**: What are the challenges involved and how can they be efficiently implemented?

> Response: *We think the reviewer here implies previously suggested methods the answer to which is given as part of the reviwer's first major concern. If the question relates to the method proposed here, then please see the answer to the 2nd point under Major concern 2.*
>
> *One thing we note is that some of the matrices involved in the sensitivity expressions can be big as the number of parameters increases. However, as long it is possible to do efficient matrix multiplication optimizing for memory, it would be possible to compute these quantities in a reasonable time. We checked computations of these methods for ~4 thousand parameters on a personal laptop and they don't require more than a few minutes. See MATLAB Livescript in the supplemental section for an implementation.*
>
> Action: *See response for bullet 2 in the Final Comments section.*

5. **Reviewer Comment**: How does one compare sensitivities of different quantities (with different units) and rank the sources of sensitivity?

Response:

*Case 1: Ranking multiple parameters with same units*

*Consider we have to rank importance of observations (called as parameters in the manuscript) with same units in influencing the estimate of fluxes for the inverse problem described in this work. LSA of fluxes with respect to each observation will result in a vector of partial derivatives (e.g. one column vector for one observation). In the case of multiple observations, it is highly likely that these derivative vectors are correlated to each other. In this situation, we want to find out (when derivative vectors are correlated) which observation is most important.*

*It is important to note that two derivative vectors with completely reverse signs can have the same norm but not be independent of each other. Therefore, they can have a completely different relationship to the MAP estimate of fluxes. It is less informative to say that the MAP estimate is equally sensitive to both observations. In fact, it may turn out that one observation "may be" completely redundant in influencing the estimate of fluxes and is just increasing the computational cost of the inverse problem.*

*We adopted a regression-based approach (see section 3.4 in the manuscript for details and also check below-given reference) to address the issue of multicollinearity among derivative vectors of the observations and rank the importance of the observations. This is also closer to the scientific goal in question. Such an approach besides taking multicollinearity into account opens up all kinds of avenues for analysis that goes with regression. Overall, we can consider this to be one potential approach for ranking the importance of observations.*

Reference:

- *Johnson, Jeff W. "A heuristic method for estimating the relative weight of predictor variables in multiple regression." Multivariate behavioral research 35.1 (2000): 1-19.*

*Case 2: Multiple parameters with different units*

*Two cases arise in this situation i.e., (1) ungrouped parameters with different units, and (2) grouped parameters with different units. In the case of ungrouped parameters, the approach as described with respect to Case 1 can be used. The second case arises when we have to rank the overall importance of a group of parameters like all observations vis-à-vis covariance parameters and prior information. This case requires that we create one column vector associated with one group that captures the overall sensitivity pattern associated with that group.*

*Inter-group derivatives can have large/small negative, positive and extreme values that depend on the unit of the quantity of interest. In this situation, we need to do some kind of normalization which does not allow negative and positive values to cancel each other and creates a pattern that is representative of the overall sensitivity of a group (in the context of this work 0-1 normalization as described in section 3.4) to the MAP estimate. A simple division by $L_2$ norm to make them unit free and summing/averaging the*

*columns up to group may not work because the summation of scaled vector derivative would result in cancellation of negative and positive values. It might be possible to use the Mahalanobis norm here that can correctly weigh the signed extreme points, but it is not easy to find the covariance/weight matrix. Therefore, we resort to a double normalization and grouping. Please see section 3.4 for a description. Once we normalize and group the derivatives, we can then rank the importance of the grouped inputs with different units by using the regression-based approach described in the context of Case 1 (mentioned above). Note this is also exemplified in the MATLAB Livescript submitted with this manuscript.*

*Also, note that ranks of parameters within a group are internally consistent as scaling does not affect the importance of parameters with respect to the MAP estimate within a group.*

*Assumptions associated with comparison*

*We have assumed that there exists a linear relationship between derivative vectors and the emissions estimate, which may or may not be true. Consequently, the ranking proposed in the manuscript does not account for the non-linear relationship between estimates of fluxes and the derivatives. If this is a concern then the strength of the nonlinear relationship among derivative vectors can be first obtained by computing the distance correlation between fluxes and the local derivatives of the parameters and then performing transformation (e.g., Box-Cox transformation) before applying the regression technique.*

*References on distance correlation:*

- *Gábor J. Székely. Maria L. Rizzo. "Brownian distance covariance." Ann. Appl. Stat. 3 (4) 1236 - 1265, December 2009. https://doi.org/10.1214/09-AOAS312*

- *Gábor J. Székely. Maria L. Rizzo. "Partial distance correlation with methods for dissimilarities." Ann. Statist. 42 (6) 2382 - 2412, December 2014. https://doi.org/10.1214/14-AOS1255*

Final Author Comment: If the reviewer has any concern about the scope then we request the reviewer to see our response to comment 3 of Dr. Rayner (first reviewer)

---

## Referee Report (RR1)

**Review of "Metrics for evaluating the "quality" in linear atmospheric inverse problems: A case study of trace gas inversion" by Yadav et al, submitted to GMD**

The manuscript describes a method to evaluate the quality of a gridded CH4 flux field obtained by solving a Bayesian linear inverse problem. The method reduces to a set of computable metrics. The "quality" of an estimate is given by the posterior distribution which quantifies the uncertainty in it. The metrics that the authors propose could attribute the uncertainty to its various causes.

These metrics, some of which are couched in terms of sensitivity analysis, are part of the verification and validation process of an inversion. The challenge lies in the computations of these sensitivities – they involve sampling and cannot be done for high-dimensional problems, where the "parameters" are discretized spatiotemporal fields (as in the case of the paper). The innovation in the paper is that the authors derive analytical expressions for these metrics, exploiting the fact that atmospheric inversions admit exact, analytical solutions. They also adapt the sensitivity metrics to the peculiarities of atmospheric inversions. Armed with these computable and inversion-relevant metrics, they seek to explain (or "evaluate the quality of") an inversion for CH4 fluxes in the Los Angeles Basin during the days of the Aliso Canyon Gas Leak (AGL), published in JGR Atmospheres, 2019 (henceforth, Yadav et al, 2019).

Yadav et al, 2019 estimated CH4 fluxes in a spatiotemporally resolved manner. They did not explicitly include AGL, either as a known source of CH4, or as one that had to be estimated. However, they did manage to capture the enhancement of CH4 fluxes in the LA basin, as the plume was captured at some of the monitoring stations. In the current manuscript, they show that the new metrics can be computed using the inverse solution in Yadav et al, 2019. They do not illustrate how these metrics can be used, to answer a scientific question. Thus one is left wondering about the purpose of computing the metrics.

**Overall comments**

- The manuscript is somewhat carelessly written. There are unfinished sentences, missing commas, anomalous indentations and capitalizations. The authors should read over the manuscript and correct these errors – I point out some of them below.
- The derivations in the paper can be involved, but nobody said spatial statistics was easy. I redid some of the derivations – the exposition is correct and clear.
- For a paper that seeks to "evaluate the quality of …", it does precious little of it. The authors compute, tabulate and plot the metrics, but do not illustrate how they may be used to answer scientific questions. For example, could the authors use the metrics to show how they managed to capture the effect of AGL? Was it caught by one monitoring site (GRA, which was near it) or by multiple ones? This may be part of the reason why one of the reviewers wonders if this is a new way of solving the inverse problem (it is not).

- On the whole, a useful paper, as it provides scalable and computable forms of the metrics. An illustration of how these metrics could be used (e.g., to answer a scientific question, to ensure correctness of solution, to interpret results or to resolve numerical issues) would be necessary to justify why these metrics are needed in the first place – and this illustration is missing. (As I will show below, this may not be very hard to do)
- I look forward to the manuscript appearing in GMD, once its minor blemishes are fixed.

**Typos and grammar (incomplete list; please read the paper carefully and fix it)**

- Line 48: "…but also admits closed form solutions". Admits, plural
- Line 49: "As we have limited knowledge …." The sentence ends before it is completed – it's just an adverbial clause.
- Line 110: "In inversions that assimilate … " (not assimilates)
- Line 119: "Note *that* sometimes …"
- Line 132, Eq 2, denominator: Should it not be $A_F \cup A_G$ rather than $A_F \cup A_F$? Also, a comma after the equation.
- Line 133: "where for any …" No indentation and start with lowercase, after the comma in Eq 2
- Line 205: "subtract a covariate by its mean …" should be "subtract the mean from the covariate …."
- Line 329: Do not use "doesn't" – it is conversational. Use "does not"
- Line 351: "where …." – no indentation and start with lowercase. Also, should have a comma after Eq. 39
- Line 365: "other than the variance based …" should be variance-based (hyphen missing)
- Line 410: "After which …" Reformulate the sentence, as it seems to be a continuation of the previous one (or merge it with the previous one)
- Line 482: "…whereas opposite …." Should be "…whereas *the* opposite …"
- Eq 13: Define A, $\Psi$, $\Omega$
- Fig 4: Subfigures A and C are sensitivities plotted over space. At what point in time were they computed?
- In Sec 4, could you describe how Q, R and X are modeled? Are they diagonal matrices? How many free parameters (to be estimated from data) do they contain? What is in X i.e., what are its columns? These are all in Yadav et al, 2019, but should be repeated here.

**Technical questions**
- Section 3 talks about Jensen-Shannon distance and GSA, but these are never used in Sec 4 (either calculated or used to illustrate a point). However, they are implemented in the released code. Since these concepts are not needed to understand the paper, move them into the Appendix?
- One question that springs to mind is how the AGL flux was estimated (rather, under-estimated), without including in the inverse model. It was definitely measured at one of the monitoring stations. The authors mentions that GRA, nearest AGL, was the most

important monitoring station (Table 1). Could the sensitivity metrics that the authors compute answer the following questions:

- o Could GRA measure AGL? Does its STAD (in Fig 3) extend that far? Given the sensitivities in Fig 4 (A), does the inversion conflate AGL with the CH4 fluxes in the vicinity of GRA, as obtained (as a prior) from CALGEM? Is the conflation of AGL with the local CALGEM fluxes the reason why it is an important station (it changes the local fluxes by a huge amount, compared to CALGEM values)?
- o Consider the premise that the AGL plume was detected at multiple monitoring stations e.g., assume that easterly winds along the San Fernando valley blew the plume eastwards over CIT, ONT and BND. ONT is an important station (Table 1). Is the importance of ONT due to the Puente Hills landfill and not AGL? Is this shown by the STAD in Fig 3 which excludes AGL? Does the low importance of CIT, lack of any CH4 sources nearby and the small sensitivity footprints negate the premise that the estimated AGL leak was informed by the easterly monitoring stations?
- o Consider the premise that the plume blew southwards, towards USC, COM and IRV. COM and IRV are important stations, USC is not. Using sensitivity footprints and STAD, can we negate the idea that southerly monitoring stations contributed to the estimation of AGL?

- The small sensitivities $\partial s/\partial z$ in Fig 4 (C) around BND are distributed everywhere except due north of it. The forward operator in Fig 4 (D) around BND is non-zero only north of it, and is headed straight into the hills of Angeles National Forest. How is it possible to have non-zero sensitivities around BND at locations where the forward operator is zero? Alternatively, since BND can only sense fluxes north of it (Fig 4(D)), how come it influence flux estimates all around it (Fig 4(C))?

---

## Author Response (AR2)

**Reviewer 1**

Dear Reviewer:

We thank the reviewer for his thorough review of our manuscript. We respond to reviewers' comments sequentially. However, we follow a separate numbering sequence for clarity for the Overall comments and Detailed comments sections. The Response label identifies the changes made in light of the reviewer's comments.

**Overall Comments**

1. **Reviewer Comment:** The manuscript presents some very interesting analyses on evaluating inversions. I think the authors have tried to address most previous reviewers' concerns, and I agree with the author's decision to submit this as a technical note. The authors have provided additional details to improve the manuscript compared to the first version. I appreciated the IAOMI and can see the value of this in satellite-based inversions, given the much higher potential for overlapping information. I was surprised that the authors didn't mention this, given their affiliations.

   *Response:* We appreciate the reviewers' comments about our effort. In light of the crucial observation of the reviewer about overlapping information, we have now included the text about the utility of IAOMI for identifying the magnitude of the shared information in the measurements in the revised manuscript (please see lines 99-100).

2. **Reviewer Comment:** However, In its current form this manuscript is extremely difficult to read. I recommend this being published in GMD, if the authors are able to make this manuscript more coherent and expand on the discussion of the figures presented in Figs.4 -6.

   *Response:* We have made extensive corrections to the write-up in the manuscript and included more text explaining the evaluation of the figures in the revised manuscript.

3. **Reviewer Comment:** Finally, there are several typographical errors in the manuscript, which is somewhat common while providing a revised version. I have highlighted a couple, but I trust a proofread to fix any remaining errors. Below I provide some suggestions to help improve the manuscript.

   *Response:* Please see our response to point 2 under the **Overall Comments** section.

**Detailed Comments**

1. **Reviewer Comment:** Lines 31-41: this paragraph comes off as jargon heavy and assumed that readers know what all the terms that are mentioned are (e.g., polynomial Chas expansion, Sobol's method etc). The authors should provide some clarifications or delete this paragraph.

*Response:* Goal of this paragraph is to provide a literature review of the most pproiment previous sentivity analyses approaches and why we need approaches that are specifically catered towards linear atmospheric inversion. We included this paragraph based on the suggestion of the earlier reviewer and provided the reference for the methods mentioned in lines 31-53. During that review round, we were not sure of including texts in lines 31-53 as it has little to do with trace gas inversions. In the revised manuscript, we have moved the text in lines 31-53 to the appendix (appendix A). Note that we are aware of the intricacies of the methods mentioned in lines 31-53.

2. **Reviewer Comment:** Line 46: Define "analytical closed-form solutions"

   *Response:* In the revised manuscript, we have included the text expanding on the meaning of the analytical closed-form solutions (please see lines 531-532 in appendix A).

3. **Reviewer Comment:** Eq 2: define what the summation is over. For instance, it could be space, time or both. This information is somewhat implicit, but I would appreciate if the authors would state this.

   *Response:* In the manuscript's construct, the summation is primarily over space for a given period (please see line 85). However, it can be both space and time. It depends on how an user is constructing the forward operator or footprint for the problem.

4. **Reviewer Comment:** Line 129. IAOMI not IOAMI right? The authors use IOAMI several times in the manuscript.

   *Response:* This is an error from our part. It should be IAOMI; we have fixed this in the revised manuscript.

5. **Reviewer Comment:** Line 139: More jargon. You could rephrase as "Jaccard similarity index which describes… "Similarly for the Ruzicka index.

   *Response:* Right. We have made the suggested corrections (please see lines 95-97). Please note that we have mentioned these names to provide connection between our proposed index and the established similarity indices in literature.

6. **Reviewer Comment:** Line 185: This is an example of how this could be particularly useful for satellite data.

   *Response:* We have now included the suggestion as text in the manuscript (please see lines 131-132).

7. **Reviewer Comment:** Lines 198-208: The authors do a good job at describing the terms in eq 9 and 10 but I think could a line or two about interpreting it. For instance, in both eqs the first term describes the observational constraint while the second term describes prior info (in eq 9) and information about fluxes (through X in eq 10). Also mention why we need the second terms in both equations.

*Response:* Following reviewer's suggestion, we have now provided a thorough description of equations 9 and 10 in the revised manuscript (please see lines 156-158).

8. **Reviewer Comment:** Line 215: Reads a little awkward. I'm guessing they are saying Beta needs to be estimated? Clarify.

   *Response:* Yes, Beta needs to be estimated, and we have clarified this in a simpler language in the revised manuscript (please see lines 158-160).

9. **Reviewer Comment:** Line 225: What does "entry" mean?

   *Response:* Entry in the context of this text implies each element in a matrix. This is a standard notation concerning matrices and their components. In the revised manuscript, we have mentioned that the entry indicates each element with the notation $ij$ in the matrix (please see lines 170-171).

10. **Reviewer Comment:** Line 319: a great time to remind your readers what GSA means (even if it has been defined earlier).

    *Response:* Incorporated. Please see line 264.

11. **Reviewer Comment:** Line 328: Define DGSM

    *Response:* Please see lines 271-272 and appendix A.

12. **Reviewer Comment:** Line 375: missing word "for" between accounted and through?

    *Response:* Right. We have incorporated it (please see line 312). Please note that we have extensively corrected the writeup in the manuscript.

13. **Reviewer Comment:** Fig 3. This is an interesting figure but perhaps also misleading. Obviously all gridcells in the shaded region for a given observation don't contain equal information. I would request the authors to provide a complimentary figure showing the footprints.

    *Response:* We have now included the figure of the footprints as requested by the reviewer. Please see Figure 4, its caption. It is referenced in line 384.

**Reviewer 2**

Dear Reviewer:

   We thank the reviewer for his thorough review of our manuscript. Our responses to reviewers' comments are given in sequential order. However, for clarity, we follow separate numbering sequences for the overall and specific comments sections. The Response label identifies the changes made considering the reviewer's comments.

1. The manuscript describes a method to evaluate the quality of a gridded CH4 flux field obtained by solving a Bayesian linear inverse problem. The method reduces to a set of computable metrics. The "quality" of an estimate is given by the posterior distribution which quantifies the uncertainty in it. The metrics that the authors propose could attribute the uncertainty to its various causes.

   *Response:* No response is required.

2. These metrics, some of which are couched in terms of sensitivity analysis, are part of the verification and validation process of an inversion. The challenge lies in the computations of these sensitivities – they involve sampling and cannot be done for high-dimensional problems, where the "parameters" are discretized spatiotemporal fields (as in the case of the paper). The innovation in the paper is that the authors derive analytical expressions for these metrics, exploiting the fact that atmospheric inversions admit exact, analytical solutions. They also adapt the sensitivity metrics to the peculiarities of atmospheric inversions. Armed with these computable and inversion-relevant metrics, they seek to explain (or "evaluate the quality of") an inversion for CH4 fluxes in the Los Angeles Basin during the days of the Aliso Canyon Gas Leak (AGL), published in JGR Atmospheres, 2019 (henceforth, Yadav et al, 2019).

   *Response:* No response is required.

3. Yadav et al, 2019 estimated CH4 fluxes in a spatiotemporally resolved manner. They did not explicitly include AGL, either as a known source of CH4, or as one that had to be estimated. However, they did manage to capture the enhancement of CH4 fluxes in the LA basin, as the plume was captured at some of the monitoring stations. In the current manuscript, they show that the new metrics can be computed using the inverse solution in Yadav et al, 2019. They do not illustrate how these metrics can be used, to answer a scientific question. Thus one is left wondering about the purpose of computing the metrics.

*Response*: We think AGL (in the reviewer's comment) means Aliso Canyon Leak (ACL) . We have premised all our responses that are associated with AGL by assuming that it stands for Aliso Canyon Leak (ACL). Based on suggestion of both the reviewers we have now included text explaining the utility of the proposed metrics in improving inversions (please see lines 422-438).

**Overall comments**

1. **Reviewer Comment:** The manuscript is somewhat carelessly written. There are unfinished sentences, missing commas, anomalous indentations and capitalizations. The authors should read over the manuscript and correct these errors – I point out some of them below.

   *Response:* We have made extensive corrections to the write-up in the manuscript and removed all the grammatical inconsistencies.

2. **Reviewer Comment:** The derivations in the paper can be involved, but nobody said spatial statistics was easy. I redid some of the derivations – the exposition is correct and clear.

   *Response:* We thank the reviewer for verifying our derivations.

3. **Reviewer Comment:** For a paper that seeks to "evaluate the quality of …", it does precious little of it. The authors compute, tabulate and plot the metrics, but do not illustrate how they may be used to answer scientific questions. For example, could the authors use the metrics to show how they managed to capture the effect of AGL? Was it caught by one monitoring site (GRA, which was near it) or by multiple ones? This may be part of the reason why one of the reviewers wonders if this is a new way of solving the inverse problem (it is not).

   *Response:* We think by AGL, the reviewer implies Aliso Canyon Leak (ACL).

   The leak signal was primarily observed at the GRA site. Its impact was not distinguishable at other sites. We can use the metrics proposed in the manuscript to identify sites that could capture the effect of the Aliso Canyon leak by looking at the sensitivity of fluxes to observations. In the revised manuscript, we have included text about the approach that can be adopted to identify impact of the signals produced by the Aliso Canyon leak in observations (please see lines 398-410).

4. **Reviewer Comment:** On the whole, a useful paper, as it provides scalable and computable forms of the metrics. An illustration of how these metrics could be used (e.g., to answer a scientific question, to ensure correctness of solution, to interpret results or to resolve

numerical issues) would be necessary to justify why these metrics are needed in the first place – and this illustration is missing. (As I will show below, this may not be very hard to do)

*Response:* We thank the reviewer for elucidating this point on utility. We have included more text in section 3.2 explaining the utility of each metric.

5. I look forward to the manuscript appearing in GMD, once its minor blemishes are fixed.

*Response:* We thank the reviewer for recognizing our contribution. In the revised manuscript, we have fixed all the issues raised by the reviewer.

**Typos and grammar (incomplete list; please read the paper carefully and fix it)**

1. **Reviewer Comment:** Line 48: "…but also admits closed form solutions". Admits, plural

   *Response:* This is an error from our part. We have now extensively corrected the write-up of the previously submitted manuscript (please see appendix A line 532).

2. **Reviewer Comment:** Line 49: "As we have limited knowledge …." The sentence ends before it is completed – it's just an adverbial clause.

   *Response:* Please see the answer to comment 1 under the section Typos and grammar (please see appendix A lines 530-534).

3. **Reviewer Comment:** Line 110: "In inversions that assimilate … "(not assimilates)

   *Response:* Incorporated; Please see the answer to comment 1 under the section Typos and grammar (please see line 68).

4. **Reviewer Comment:** Line 119: "Note *that* sometimes …"

   *Response:* Incorporated; Please see the answer to comment 1 under the section Typos and grammar (please see line 76).

5. **Reviewer Comment:** Line 132, Eq 2, denominator: Should it not be $A_F \cup A_G$ rather than $A_F \cup A_F$? Also, a comma after the equation.

   *Response:* It is an error from our part. We have fixed the typo in the equation 2 in the revised manuscript (please see line 89). We have also added comma after all such instances in the revised manuscript.

6. **Reviewer Comment:** Line 133: "where for any …" No indentation and start with lowercase, after the comma in Eq 2

   *Response:* Fixed the indention and changed the text to lowercase in the revised manuscript (please see line 90).

7. **Reviewer Comment:** Line 205: "subtract a covariate by its mean …" should be "subtract the mean from the covariate …."

   *Response:* Right. Fixed in the revised manuscript (please see line 147).

8. **Reviewer Comment:** Line 329: Do not use "doesn't" – it is conversational. Use "does not"

   *Response:* This is an error from our part. We have fixed it in the revised manuscript (please see line 273).

9. **Reviewer Comment:** Line 351: "where …." – no indentation and start with lowercase. Also, should have a comma after Eq. 39

   *Response:* Incorporated; please see the answer to comment 1 under the section Typos and grammar (please see lines 282-283).

10. **Reviewer Comment:** Line 365: "other than the variance based …" should be variance-based (hyphen missing)

    *Response:* Incorporated; please see the answer to comment 1 under the section Typos and grammar (please see line 302).

11. **Reviewer Comment:** Line 410: "After which …" Reformulate the sentence, as it seems to be a continuation of the previous one (or merge it with the previous one)

    *Response:* Incorporated; please see the answer to comment one under the section Typos and grammar (please see line 344-345).

12. **Reviewer Comment:** Line 482: "…whereas opposite …." Should be "…whereas *the* opposite …"

    *Response:* We have restructured this part in the revised manuscript. Please see lines 421-438.

13. **Reviewer Comment:** Eq 13: Define A, Ψ, Ω

    *Response:* These terms are defined in Equation 14. We have now referenced it in the Equation 13 description (please see line 163).

14. **Reviewer Comment:** Fig 4: Subfigures A and C are sensitivities plotted over space. At what point in time were they computed?

    *Response:* These figures are associated with the period of the case study, and we have now mentioned this in the caption of the figure 5.

15. **Reviewer Comment:** In Sec 4, could you describe how Q, R and X are modeled? Are they diagonal matrices? How many free parameters (to be estimated from data) do they contain? What is in X i.e., what are its columns? These are all in Yadav et al, 2019, but should be repeated here.

    *Response:* We have now included details about how Q, R and X are modeled in the inversion (please see lines 361-363). These details also appear in the code submitted as part of this manuscript.

**Technical questions**

1. **Reviewer Comment:** Section 3 talks about Jensen-Shannon distance and GSA, but these are never used in Sec 4 (either calculated or used to illustrate a point). However, they are implemented in the released code. Since these concepts are not needed to understand the paper, move them into the Appendix?

    *Response:* We agree with the reviewer that these concepts are not needed to understand the manuscript. However, both the concepts have significant implication in the general linear atmospheric inversion evaluation framework. Dissimilarity measures like 1-IAOMI or Jensen-Shannon distances could be used to model the transport error component of the model-data error covariance. On the otherhand, GSA is crucial to get a comprehensive idea about the uncertainty apportionment. Being able to perform both LSA and GSA can provide a complete evaluation of the estimated fluxes that is otherwise sufficient but not complete. In light of the reviewer's comment, we have now moved the JSD discussion (being another approach to dissimilarity) in the appendix but we kept the GSA as mentioned above. We believe that it can provide users a continuity and completeness of the approaches.

2. **Reviewer Comment:** One question that springs to mind is how the AGL flux was estimated (rather, underestimated), without including in the inverse model. It was

definitely measured at one of the monitoring stations. The authors mentions that GRA, nearest AGL, was the most important monitoring station (Table 1). Could the sensitivity metrics that the authors compute answer the following questions:

- **Reviewer Comment:** Could GRA measure AGL? Does its STAD (in Fig 3) extend that far?

*Response* : The STAD of GRA just implies that compared to other sites, it is the site to see the signal of fluxes from the area identified in Figure 2. However, this does not mean that it is sensitive to the fluxes in the area for every period. STAD does not represent the coverage of the network i.e., regions of emissions constrained by observations. These regions are shorter than STAD. They are either obtained before performing an inversion by identifying areas of continuous spatiotemporal coverage with certain percentage of magnitude as provided by atmospheric transport or by assessing the model resolution after performing an inversion (for an explanation, see Yadav et al., 2019). We have identified this region by the yellow outline in Figure 2 of the manuscript.

Overall, STAD for each site indicates regions of emissions in the LA basin that contribute most to the enhancement signal observed at a site. Therefore, we can associate the change in emissions to the specific area in the basin where reductions or increases in emissions are likely to have occurred.

- **Reviewer Comment:** Given the sensitivities in Fig 4 (A), does the inversion conflate AGL with the CH4 fluxes in the vicinity of GRA, as obtained (as a prior) from CALGEM

*Response:* Inversion does conflate emissions from the Aliso Canyon Leak with emissions from other sources (especially those from Sunshine Landfill) that are close to the leak's site. Still, we looked at the fluxes at the basin scale where the impact of the Aliso Canyon Leak is visible. Note that we only inferred fluxes from the Aliso Canyon Leak by comparing it to the baseline emissions obtained from the inversion for the last 4-day period before the period for inversion mentioned in this study. Finally, inversion based on enhancements is adjusting the prior fluxes from CALGEM; therefore, the impact of CALGEM fluxes should be minimal.

- **Reviewer Comment:** Is the conflation of AGL with the local CALGEM fluxes the reason why it is an important station (it changes the local fluxes by a huge amount, compared to CALGEM values)?

*Response:* No. In this inversion, as in a previous study by Yadav et al. 2019, the CALGEM emissions remain constant across periods (from 2015-2016, note not described in this study). Therefore, the conflations in the emissions due to the prior emissions should remain constant, and any adjustment in the magnitude of emissions is just due to enhancements in

conjunction with atmospheric transport. GRA was the most important site in quantifying Aliso leak due to the enhancement observed at this site. This effect can also be shown by computing the standard deviation of the measurements (concentrations, not enhancements) collected at the cadence of less than a minute and aggregated at a resolution of 1 hour, the time series of which can be plotted for the entire duration (2015-2016) for which data was available.

- **Reviewer Comment:** Could the sensitivity metrics that the authors compute answer the above-mentioned questions?

*Response:* The points raised by the reviewer are mostly related to the assumptions or hypotheses of the study. The proposed metrics cannot directly answer them, but they can certainly help improve the understanding of all the concerns raised by the reviewer. Thus, (1) STAD in conjunction with sensitivity maps can help in understanding the spatial domain from which emission signal is being captured over a time period of interest, (2) conflation of fluxes can be understood by studying the sensitivity of fluxes to observations and observing evolution of these sensitivities over time. Note that unlike the established network in place in LA, a denser network will have ability to resolve facility scale conflation. However, the existing network in LA was primarily designed to answer questions related to changes in basin scale fluxes not facility level changes.

3. **Reviewer Comment:** Consider the premise that the AGL plume was detected at multiple monitoring stations e.g., assume that easterly winds along the San Fernando valley blew the plume eastwards over CIT, ONT and BND. ONT is an important station (Table 1). Is the importance of ONT due to the Puente Hills landfill and not AGL? Is this shown by the STAD in Fig 3 which excludes AGL?

*Response:* Puente hills landfill has no role in determining the importance of ONT as this landfill was shut down in 2014, long before the case study presented in this work. We think ONT is important as it is downwind of all the emission sources in the LA basin, which is not the case for all other sites. As mentioned previously, the STAD of a site implies that compared to other sites, it is the site most likely to see a signal of fluxes from its STAD. This does not mean that a measurement can't be influenced by certain sporadic emission from a source. However, if this is persistent, it will eventually be part of the STAD region for a particular site.

4. **Reviewer Comment:** Does the low importance of CIT, lack of any CH4 sources nearby and the small sensitivity footprints negate the premise that the estimated AGL leak was informed by the easterly monitoring stations? o Consider the premise that the plume blew southwards, towards USC, COM and IRV. COM and IRV are important stations, USC is not. Using sensitivity footprints and STAD, can

we negate the idea that southerly monitoring stations contributed to the estimation of AGL?

*Response:* Right. The points raised by the reviewer can quickly be answered by looking at the forward operator and the sensitivity of fluxes to observations, as shown in Figure 5 in conjuction with STADs. We have only shown this as the most important observation, but a detailed analysis of del shat/del $z_i$ can answer this question. However, we do not conduct these analyses in this study as that is not the study's goal.

5. **Reviewer Comment:** The small sensitivities $\partial s)_{\partial z}$ in Fig 4 (C) around BND are distributed everywhere except due north of it. The forward operator in Fig 4 (D) around BND is non-zero only north of it and is headed straight into the hills of Angeles National Forest. How is it possible to have non-zero sensitivities around BND at locations where the forward operator is zero? Alternatively, since BND can only sense fluxes north of it (Fig 4(D)), how come it influence flux estimates all around it (Fig 4(C))?

*Response:* This happens as delshat/delz (sensitivity of fluxes to observation) obtained after an inversion can be non-zero at a location where a forward operator of observation from the nearest or most influential site is 0. This can happen because of the overlap of the forward operators from the non-influential far away sites. These delshat/delz values are generally tiny as can be seen in Figure 5 (C). Numerical precision is also one of the reasons why this occurs.